# Positive appraisal style predicts long-term stress resilience and mediates the effect of a pro-resilience intervention

Papoula Petri-Romão[1], Roberto Mediavilla [2,3,4] ✉,
Alexandra Restrepo-Henao[5,6], Lara MC Puhlmann [1,7], Matthias Zerban[1,8],
Kira F. Ahrens[9,10], Corrado Barbui[11], Ulrike Basten [12], Carmen Bayón[2,3,13,14],
Andrea Chmitorz[15], Mireia Felez-Nobrega[16,17], Bianca Kollmann[1,18], Klaus Lieb[1,19],
David McDaid[20], Kerry R. McGreevy[2,3], Maria Melchior [21],
Ainoa Muñoz-Sanjosé[2,3,13,14], Rebecca J. Neumann [9], A-La Park[20],
Michael M. Plichta[9], Marianna Purgato[11], Andreas Reif [9], Charlotte Schenk[9],
Anita Schick[22], Alexandra Sebastian[1,19], Marit Sijbrandij[23], Pierre Smith[24,25],
Oliver Tüscher [1,19,26,27], Michèle Wessa[1,18,28,29], Anke B. Witteveen [23],
Kenneth SL Yuen [1,8], Josep Maria Haro[3,17,19,30],
José Luis Ayuso-Mateos[2,3,4,31,32] & Raffael Kalisch[1,8,32]

Stress resilience is the maintenance of mental health despite adversity. Identifying factors that predict and promote good long-term mental health outcomes in stressor-exposed individuals is a first step towards developing more effective prevention programs. In two independent observational samples ($N = 132$, $N = 1034$), we find that a tendency to evaluate stressors in a realistic to slightly unrealistically positive fashion (positive appraisal style, PAS) is prospectively associated with resilient outcomes over several years. We also find that PAS is an integrative, proximal resilience factor that mediates the pro-resilience effects of other protective factors (e.g., social support). In an analysis of pre-specified exploratory outcomes of a randomized controlled trial comparing a behavioral intervention targeting a broad set of resilience factors against usual care in a sample of distressed healthcare workers ($N = 232$; trial registry: NCT04980326), we find that PAS is modifiable, with improvements in PAS mediating intervention-induced improvements in resilience. These results establish PAS as a proximal, plastic, and potentially causal resilience factor.

Mental health conditions such as anxiety, depression, or post-traumatic stress disorder are partly caused by exposure to stressors. These can include adverse life events, longer-term difficult life circumstances, or challenging life transitions[1–4]. However, not all individuals exposed to stressors develop mental health problems—a phenomenon known as stress resilience[5]. In times of multiplying global crises and a high and rising disease burden from stress-related disorders[6], this observation is raising increasing interest as an anchor point to develop new strategies in the combat for mental health[5,7]. Specifically, it is hoped that the identification of social, psychological, or biological resilience-predictive factors, along with their causal links to resilient outcomes, can inspire prevention programs for particularly vulnerable groups.

Appraisal is the evaluation of a stimulus or situation in terms of its meaning for the needs and goals of the individual. It is considered by appraisal theories as the determinant of the emotional reaction to the

stimulus/situation (e.g. refs. 8–11). Stressors are stimuli that are appraised as potential threats to one's needs or goals. Positive appraisal style theory of resilience (PASTOR)[12] builds on these concepts by claiming that a general tendency to appraise stressors in a realistic to mildly illusionary positive fashion (a 'positive appraisal style', PAS) is a key resilience factor. An overly (delusionally) positive appraisal tendency is characterized by very high levels of threat trivialization, optimism, and confidence in one's coping abilities (right side in Fig. 1). In comparison, the milder average levels of these appraisal biases that characterize a PAS (green box in the figure) still allow the individual to mount stress reactions when necessary to deal with potentially threatening situations and prevent harm. At the same time, compared to an unrealistically negative appraisal tendency[13] characterized by catastrophizing, pessimism, and perceived helplessness (left side in the figure), individuals with a PAS will be more likely to avoid unnecessary stress reactions or over-reactions. That is, their stress responses will on average be well regulated in terms of magnitude (no over-shoot in response amplitude), duration (fast recovery after stressor termination), and quality (appropriate response selection)[12,14]. Habitual positive appraisers will thereby be better protected against resource depletion and have more room for learning, exploration, creativity, restoration, and resource-building than habitual negative appraisers. In situations of adversity or longer-lasting stressor exposure, this balanced stress response profile provides an optimal cost-benefit ratio and will eventually reduce an individual's risk of developing mental health problems[12,14].

Initial studies have shown that PAS was linked with relatively better stress resilience during the first phases of the COVID-19 pandemic. This was observed cross-sectionally in 16,000 European adults[15] and in 570 mental health practitioners from various countries[16] and also prospectively over 5 weeks in 200 European adults[17] and over 6 months in 350 Dutch patients with Parkinson's Disease[18]. These studies measured PAS with an early version of the Perceived Positive Appraisal Style Scale, process-focused (PASS-process)[15]. This self-report questionnaire was developed during the beginning of the pandemic to assess cognitive processes and strategies that individuals employ under stress to produce positive appraisals, that is, to view a difficult situation more positively. Questionnaire items mainly represent different variants of positive cognitive reappraisal.

The first goal of the present work is to ask whether PASS-process scores, now obtained from a fully validated version of the instrument[19], prospectively predict resilience across much longer timeframes. This is done in data from two German longitudinal observational studies, the Mainz Resilience Project (MARP)[20,21] and the population-based Longitudinal Resilience Assessment (LORA) study on resilience to everyday modern-life stressors[22]. MARP is a sample of healthy young adults confronted with the challenges of transitioning into adulthood and pre-selected for having experienced at least three prior negative life events. Participants have reported their exposure to stressors and potential internalizing (negative mood and affective) symptoms every 3 months over 3.7 years. In LORA, the corresponding 3 monthly stressor and symptom reports have been obtained from adult participants

over 3 years. In both samples, the baseline PASS-process measurement is complemented with an additional recently validated instrument, the Perceived Positive Appraisal Style Scale, content-focused (PASS-content)[19]. In this questionnaire, individuals report to what extent they generate positive appraisals when they are under stress. This scale targets the PAS construct more directly than PASS-process because it focuses on appraisal contents, rather than on the cognitive processes that can generate these contents (see Discussion for a more in-depth distinction).

As in the initial studies using only PASS-process[15–18], resilience in these samples is defined as the maintenance or quick recovery of mental health during and after stressor exposure –or, in other words, a good long-term mental health outcome despite adversity[5]. Resilience is quantified by first regressing participants' scores for internalizing mental health problems on their stressor exposure scores. The regression line describes the normative reactivity of participants' mental health to the reported stressors in the given sample. We then express an individual participant's 'stressor reactivity' at each reporting time point by their residualized mental health problem score, that is, the distance of the mental health problem score from the normative regression line. We thus obtain a continuous stressor reactivity score (SR score), where a smaller value indicates that the participant is relatively less affected by the stressors[23], compared to the given sample. Critically, the residualization approach (see also refs. 24–29) accounts for the fact that stressor exposure can differ between individuals. Classifying an individual as resilient merely based on low raw mental health problem scores ignores that a person may show good mental health for the trivial reason that they experience less adversity[23,30]. By contrast, by normalizing for exposure, the SR score can be compared between participants with different exposure levels, such that relatively lower SR in a participant corresponds to relatively better mental health despite adversity[23]. Resilience as 'good long-term mental health despite adversity[5] is operationalized even better when a stressor-exposed individual shows low SR over longer time frames[23], such as now available in MARP and LORA.

PASTOR also claims that PAS is an integrative resilience factor that mediates the effects of other (social, psychological, or biological) resilience factors in that these factors shape the way an individual typically perceives stressors. In turn, this will determine how much an individual typically reacts to them and, eventually, how resilient they are[12,31]. Thus, PAS acts on resilience more proximally than other factors. In the four previous studies conducted during the COVID-19 pandemic[15–18], we found that negative associations between the well-established resilience factor perceived social support[32–35] and SR were mediated by PASS-process. This supports the idea that individuals who trust more in the availability of assistance from their social networks also perceive difficult situations as more easily controllable or less impactful and therefore show more resilience[15,34]. As explained, PASTOR also posits that positive appraisers show an optimal stress response profile, which eventually protects their mental health in adverse life situations[12,14]. In line with that claim, we also observed that negative associations of PASS-process with SR were mediated by a self-

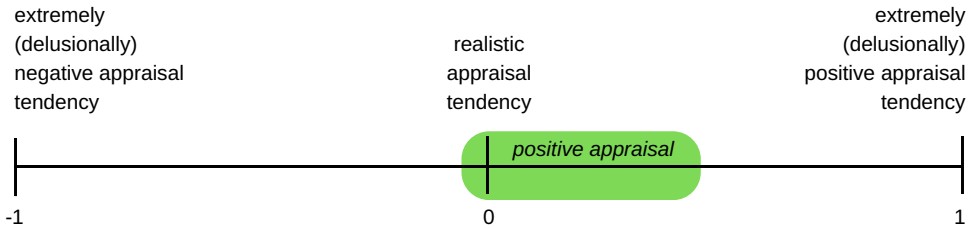

**Fig. 1 | Schematic representation of Positive Appraisal Style (PAS) according to Positive Appraisal Style Theory of Resilience (PASTOR).** PAS represents a tendency to appraise stressors in a realistic to mildly illusionary positive fashion, as approximatively represented by the green range in the figure. Adapted from Kalisch and colleagues[84] © Cambridge University Press 2015, reproduced with permission.

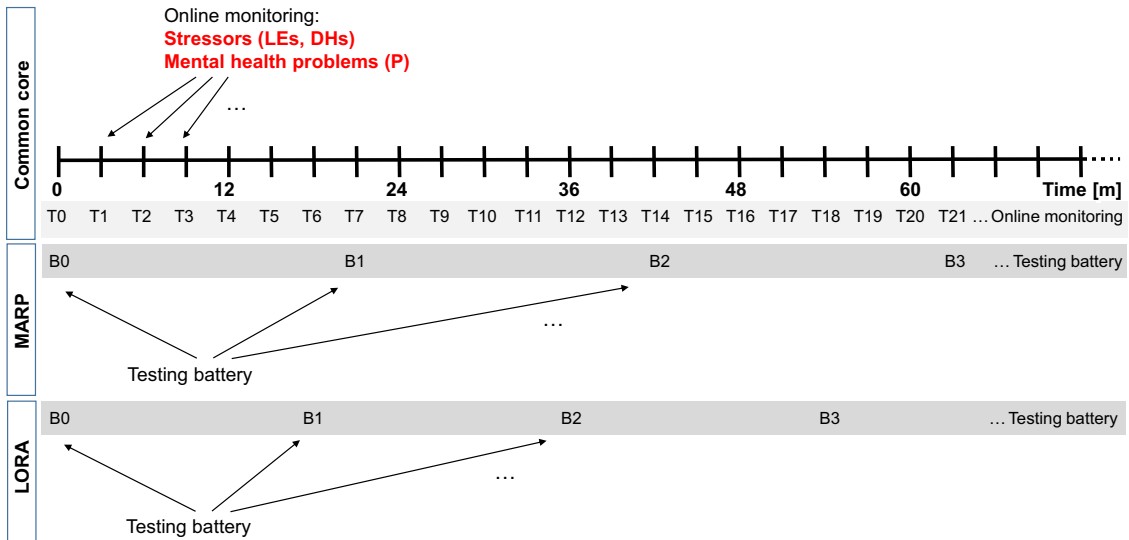

**Fig. 2 | MARP and LORA study designs.** Both studies implement the frequent stressor and mental health monitoring (FRESHMO) paradigm[23], where stressors (life events, daily hassles) and mental health problems (internalizing symptoms) are repeatedly and frequently assessed, via a 3 monthly online monitoring (T0, T1, T2, …). This monitoring scheme permits us to describe mental health changes associated with stressor exposure. In order to identify resilience factors, such as PAS, a testing battery (B0, B1, B2,…) is repeatedly administered approximately every 1.75 (MARP) and 1.5 (LORA) years.

assessment of participants' stress recovery. That is, participants with a higher PAS also reported to more easily recover from stressors and showed correspondingly lower SR[15-17]. The second goal of the present work was therefore to test the generalizability of these observations from samples primarily challenged by the pandemic to samples primarily exposed to other modern-life stressors (MARP and LORA). We also tested whether mediations are observed not only for PASS-process, but also for PASS-content.

The idea of PAS as an integrative and proximal mediator of other resilience factors implies that PAS may be enhanced even by interventions that do not target it directly. This means that interventions trying to boost other (more distal) resilience factors, or a broad set of potentially resilience-promoting mechanisms should also improve PAS (for an extended theoretical discussion, see Supplementary Note 1). Importantly, if such interventions improved PAS, this improvement should translate into better resilience outcomes (lower SR). Such a result would also hint at a causal role for PAS in resilience.

The third goal of the present work therefore was to investigate whether a mechanistically broad behavioral intervention improved PAS and resilience outcomes. To that end, we performed a pre-specified analysis of exploratory outcomes of a randomized controlled trial (RCT) testing a psychological stepped-care intervention in healthcare workers in Spain during the COVID-19 pandemic (RESPOND-RCT Spain; ClinicalTrials.gov Identifier: NCT04980326)[36]. Early analyses had indicated that healthcare workers are among the populations whose mental health was most impacted by the pandemic[37]. In this trial, stressor exposure, internalizing mental health problems, and PASS-content were assessed at four time points (baseline, peri-intervention, post-intervention, follow-up), spaced several weeks apart, both in the intervention and in the usual care arms. The first intervention step offers practices based on acceptance and commitment and mindfulness approaches that aim at modifying cognition through experiencing reality in a different way[38]; the second step explicitly targets dysfunctional behaviors and cognitions[39]. Hence, the intervention used a multi-component approach serving to initiate or enhance a broad range of potentially protective processes, making it suitable to test mediation by PAS. The primary analysis of the trial data has shown that the intervention successfully reduces internalizing symptoms[40].

In both MARP and LORA, we find a negative prospective association between PASS-content and long-term SR scores (goal 1). PASS-content also mediates the negative association between perceived social support and SR, and its negative association with SR is in turn mediated by perceived good stress recovery in both samples (goal 2). Last, results from RESPOND-RCT Spain reveal that the intervention reduces SR and enhances PASS-content. Baseline PASS-content shows a negative prospective association with SR at later time points and, most importantly, the increase in PASS-content from baseline to post-intervention prospectively and strongly mediates the decrease in SR from baseline to follow-up, in line with a possible causal contribution of PAS to resilience (goal 3).

## Results

### Observational discovery sample: MARP

In the ongoing MARP study, a mixed laboratory and online battery assessing potential resilience factors (such as PAS) and other mental health predictors is administered approximately every 1.75 years (time points B0, B1, B2, … in Fig. 2). Every 3 months, starting at the time of the first battery (time points T0 = B0, then T1, T2, T3, …), exposure to both macrostressors (life events) and microstressors (daily hassles) as well as the magnitude of potential internalizing mental health problems (General Health Questionnaire-28, GHQ-28)[41] are monitored via online self-report. Of the 200 participants included at study baseline (B0/T0) between July 2016 and March 2019, $N = 132$ could be used for longitudinal analyses that covered online monitoring until the time of the third battery administration at B2. The average B0-B2 interval was 3.7 years (range 2.8–4.8 years, see Methods). At baseline, these participants had a mean age of 19.2 years (SD = 0.8), 83 (62.9%) were female, 92 (69.69%) were university students. Average baseline scores on the mental health instrument were 21.0 (SD = 9.0, possible range 0–84). A recommended screening cut-off for the GHQ-28 is 23/24[42], indicating that some mental problem load was present in this sample already at inclusion. This is consistent with the selection criterion of having experienced at least three negative life events and with participants being confronted with the challenges of transiting into adulthood. For further sample characteristics, see Supplementary Table 1.

Over the 3.7 years, the most frequent life events reported by participants at the occasion of the online monitorings (T1, T2, T3, …)

were in the categories of 'other impactful event' (such as exams, accidents, natural disasters, or armed conflicts; mean ($M$) = 0.5 (SD = 0.5) times per 3 monthly monitoring time window), constant arguments between family members ($M$ = 0.3, SD = 0.5), and serious illness, accident or diagnosis of oneself or a close family member ($M$ = 0.2, SD = 0.4). The life events with the highest severity rating (from 1 to 5) were serious arguments with boyfriend/girlfriend or spouse ($M$ = 4.0, SD = 0.9), break-up with girlfriend/boyfriend/spouse ($M$ = 4.0, SD = 1.0), and death of a beloved pet ($M$ = 3.9, SD = 1.0). The most frequently reported daily hassles in the sample were household management ($M$ = 4.5 (SD = 1.9) days per past week at each 3 monthly monitoring), high performance demand or workload at work/school/university ($M$ = 4.4, SD = 2.0), and commuting ($M$ = 3.9, SD = 1.8). The daily hassles rated as most severe were bad news ($M$ = 3.7, SD = 1.2), high performance demand or workload at work/school/university ($M$ = 3.7, SD = 1.0), and performance situations at work/school/university (e.g., exam) ($M$ = 3.6, SD = 1.1). See Supplementary Tables 2a, b for further details on stressor exposure.

A stressor exposure score E, aggregating life event and daily hassle counts, explained 28.5% of variance in the mental health problem score P (GHQ-28) (conditional $R^2$; 4.8 % marginal $R^2$) in a linear mixed model across all online monitoring time points and participants (see Methods). This allowed us to calculate SR scores to obtain an inverse outcome-based measure of resilience.

Intra-class correlations (ICCs) between battery time points B0 and B1 for PASS-content (ICC = 0.75, 95% CI [0.68,0.81], $F$(199) = 7.1 $p < 0.00001$) and PASS-process (ICC = 0.58, 95% CI [0.48,0.66], $F$(1190) = 3.7, $p < 0.00001$) indicated long-term stability of the two PAS measures. They were also highly correlated with each other at both battery assessments (B0: $r$(175) = 0.56, $p < 0.00001$; B1: $r$(112) = 0.60, $p < 0.00001$), congruent with them indexing the same psychological construct.

Controlling for age, gender, childhood trauma as well as smoking, PASS-content was prospectively negatively associated with the SR scores from B0 to B2, that is, with average SR calculated using the online monitoring data starting with the first post-baseline time point, placed 3 months after B0 (T1 in Fig. 2), until the time point concordant with B2, approximately 3.7 years after B0. After adjustment, PASS-content explained 22% of the variance in SR. See Table 1. For PASS-content, significant negative prospective relationships were also found at shorter time scales (predicting SR in the B0-B1 interval or at the first three monitoring time points (9 months) after B0 from scores at B0). Analogously, PASS-content assessed at B1 significantly negatively predicted SR in the B1-B2 interval or at the first three monitoring time points after B1 from scores at B1. Note, however, that the full model for the prediction of SR in the B1-B2 interval was not significant. PASS-process only predicted SR in the short intervals of 9 months after B0 and B1, but in both cases the models did not reach significance. In the subsample of the most stressor-exposed participants[23] (top two terciles of mean E between B0 and B2), PASS-content was predictive of SR in all time intervals but one, whereas models with PASS-process were not statistically significant (Supplementary Table 3). These observations indicate that PAS as assessed with PASS-content is a resilience factor, as hypothesized.

Next to PASS-content, perceived social support and perceived good stress recovery also showed covariate-controlled negative prospective associations with SR (Supplementary Tables 4 and 5), although the association with SR was not significant for social support in the B0-B1 interval. The prospective associations of social support(B0) with SR(B0-B2) were mediated by PASS-content (B0), but not PASS-process(B0). The associations of PASS-content(B0) and PASS-process(B0) with SR(B0-B2) were in turn mediated by good stress recovery (B0) (Fig. 3). Note, however, that these mediation analyses were underpowered (see Methods) and are only reported for discovery purposes here.

Overall, these findings suggest a link between positive appraisal as measured with the instrument directly assessing appraisal contents (PASS-content) and resilience, potentially via optimization of stress responses (better stress recovery), and they indicate that positive appraisal style may be a more proximal resilience factor than social support.

Note that, although the MARP study was designed among others to provide data to test PASTOR[20], the exact testing methods were not prespecified at the start of the study and the PASS instruments and the analytical strategies were developed during study conduct based on experiences with MARP and other studies (see Introduction)[12,15–17,19,43]. Hence, all analyses of MARP data reported here are of exploratory nature and require independent replication. The following analyses of the LORA sample serve this purpose and were set up to match the analyses of the MARP sample as closely as possible. Because PASS-process had shown associations with resilience in earlier studies (see Introduction), we also tested PASS-process in LORA, reporting results descriptively only in tables and figures.

### Observational replication sample: LORA

In the ongoing LORA study, a resilience factor battery partly overlapping with the one used in MARP is administered approximately every 1.5 years (B0, B1, B2, …), and online monitoring using the same instruments as MARP also occurs every 3 months (T0, T1, T2, …) (see Fig. 2). Of the 1,091 participants included at baseline (B0/T0) between October 2016 and July 2019, $N$ = 1,034 could be used for the longitudinal analyses until B2, that is, approximately 3 years after baseline. At baseline, these participants had a mean age of 28.8 years (SD = 8.0) and were mostly female ($n$ = 686, 66.3%). Five hundred seventy-six (52.4%) were university students, 466 (45.1%) had a university education, and 463 (44.8%) were in employment. Average baseline scores on the mental health instrument (GHQ-28) were 16.5 (SD = 7.7), that is, lower than in MARP, in line with LORA participants not being preselected for risk (for further sample characteristics, see Supplementary Table 6).

Over the 3 year period, the most frequent life events reported by participants at the online monitoring were 'other impactful event' ($M$ = 0.3 (SD = 0.4) times per 3-monthly monitoring time window), constant arguments between family members ($M$ = 0.4, SD = 3), and serious arguments with boyfriend/girlfriend/spouse ($M$ = 0.2, SD = 0.4). The life events rated as most severe were break-up with boyfriend/girlfriend/spouse ($M$ = 2.9, SD = 1.0), difficult pregnancy or miscarriage of partner or oneself ($M$ = 2.8, SD = 1.3), and serious argument with boyfriend/girlfriend/spouse ($M$ = 2.6, SD = 1.0). The most frequently reported daily hassles were household management ($M$ = 4.8 days (SD = 2.0) per past week at each 3-monthly monitoring), commuting ($M$ = 4.1, SD = 1.5), and high performance demand or workload at work/school/university ($M$ = 4.0, SD = 2.0). The daily hassles rated as most severe were conflict or disagreement with close persons ($M$ = 1.7, SD = 1.2), high performance demand ($M$ = 1.6, SD = 1.1), and time pressure ($M$ = 1.6, SD = 1.1). From the severity ratings it becomes apparent that LORA participants perceived their stressors as less burdensome than MARP participants, presumably reflecting that the LORA sample was not enriched for individuals in critical life phases (for further details, see Supplementary Tables 7a, b).

The aggregated stressor exposure score E explained 35.8% of variance (conditional $R^2$) in the mental health problem score P in a mixed linear model across all time points and participants (marginal $R^2$ = 12.2%, including fixed effects only).

ICCs between B0 and B1 of the PAS instruments were similar to MARP (PASS-content: ICC = 0.72, 95% CI [0.69,0.75], $F$(1190) = 6.2, $p < 0.00001$); PASS-process: ICC = 0.57, 95% CI [0.53,0.60], $F$(1190) = 3.6, $p < 0.00001$, and both instruments were again highly correlated at each battery assessment (B0: $r$(1150) = 0.54, $p < 0.0001$; B1: $r$(873) = 0.55, $p < 0.0001$).

**Table 1 | MARP: Prediction of stressor reactivity (SR) by PASS-content and PASS-process, controlling for baseline (B0) covariates**

| Predictor time point (battery) | B0 | | | | | | B1 | | | |
|---|---|---|---|---|---|---|---|---|---|---|
| Outcome interval (SR score) | B0-B2 (~3.7 years) | | B0-B1 (~1.9 years) | | 3 monitorings post B0 (~9 m) | | B1-B2 (~1.6 years) | | 3 monitorings post B1 (~9 m) | |
| | PASS-content | PASS-process | PASS-content | PASS-process | PASS-content | PASS-process | PASS-content | PASS-process | PASS-content | PASS-process |
| PAS | **−0.308** | −0.149 | **−0.363** | −0.161 | **−0.277** | **−0.186** | **−0.331** | −0.242 | −0.329 | **−0.228** |
| | **(−0.468, −0.149)** | (−0.319, 0.021) | **(−0.530, −0.195)** | (−0.341, 0.020) | **(−0.443, −0.110)** | **(−0.363, −0.010)** | **(−0.571, −0.090)** | (−0.483, −0.001) | (−0.547, −0.111) | **(−0.447, −0.009)** |
| | **p = 0.0003** | p = 0.089 | **p = 0.00005** | p = 0.084 | **p = 0.002** | **p = 0.041** | **p = 0.009** | p = 0.053 | p = 0.005 | **p = 0.045** |
| Age | −0.142 | −0.136 | −0.146 | −0.132 | −0.083 | −0.068 | −0.089 | −0.028 | −0.049 | −0.015 |
| | (−0.333, 0.050) | (−0.336, 0.063) | (−0.346, 0.055) | (−0.347, 0.083) | (−0.296, 0.130) | (−0.287, 0.151) | (−0.363, 0.186) | (−0.315, 0.258) | (−0.300, 0.201) | (−0.276, 0.246) |
| | p = 0.151 | p = 0.184 | p = 0.157 | p = 0.232 | p = 0.446 | p = 0.544 | p = 0.529 | p = 0.849 | p = 0.700 | p = 0.914 |
| Gender | **0.425** | **0.358** | **0.384** | 0.324 | 0.297 | 0.212 | 0.305 | 0.341 | 0.235 | 0.242 |
| | **(0.105, 0.745)** | **(0.022, 0.694)** | **(0.049, 0.718)** | (−0.036, 0.685) | (−0.041, 0.636) | (−0.141, 0.564) | (−0.143, 0.753) | (−0.129, 0.810) | (−0.162, 0.633) | (−0.175, 0.659) |
| | **p = 0.011** | **p = 0.039** | **p = 0.027** | p = 0.081 | p = 0.089 | p = 0.242 | p = 0.186 | p = 0.160 | p = 0.250 | p = 0.259 |
| Childhood trauma | **0.218** | 0.408 | **0.223** | 0.442 | 0.040 | 0.177 | 0.132 | 0.104 | 0.241 | 0.240 |
| | **(−0.254, 0.690)** | (−0.073, 0.889) | **(−0.264, 0.710)** | (−0.067, 0.952) | (−0.519, 0.599) | (−0.397, 0.751) | (−0.578, 0.842) | (−0.638, 0.847) | (−0.398, 0.881) | (−0.429, 0.909) |
| | **p = 0.368** | p = 0.099 | **p = 0.372** | p = 0.092 | p = 0.890 | p = 0.547 | p = 0.717 | p = 0.784 | p = 0.462 | p = 0.485 |
| Smoking | 0.047 | 0.051 | 0.045 | 0.052 | 0.040 | 0.048 | 0.030 | 0.051 | 0.027 | 0.050 |
| | (−0.033, 0.128) | (−0.033, 0.134) | (−0.038, 0.128) | (−0.037, 0.140) | (−0.046, 0.126) | (−0.040, 0.136) | (−0.072, 0.132) | (−0.053, 0.156) | (−0.066, 0.121) | (−0.046, 0.146) |
| | p = 0.252 | p = 0.236 | p = 0.294 | p = 0.255 | p = 0.363 | p = 0.290 | p = 0.566 | p = 0.341 | p = 0.570 | p = 0.311 |
| Number of assessments | **−0.142** | −0.190 | 0.127 | 0.112 | | | −0.065 | −0.083 | | |
| | **(−0.278, −0.005)** | (−0.333, −0.047) | (−0.131, 0.385) | (−0.169, 0.392) | | | (−0.246, 0.116) | (−0.272, 0.107) | | |
| | **p = 0.044** | p = 0.011 | p = 0.338 | p = 0.437 | | | p = 0.485 | p = 0.395 | | |
| Constant | 3.506 | 3.883 | 0.984 | 0.631 | 1.019 | 0.668 | 1.417 | 0.336 | 0.220 | −0.489 |
| | (−0.771, 7.784) | (−0.575, 8.341) | (−3.327, 5.294) | (−3.956, 5.217) | (−3.227, 5.265) | (−3.702, 5.038) | (−4.298, 7.131) | (−5.624, 6.295) | (−4.754, 5.195) | (−5.675, 4.698) |
| | p = 0.111 | p = 0.091 | p = 0.656 | p = 0.788 | p = 0.640 | p = 0.766 | p = 0.629 | p = 0.913 | p = 0.932 | p = 0.855 |
| Observations (n) | **132** | **131** | **128** | **126** | **114** | 113 | 83 | 80 | **85** | 83 |
| R² | **0.224** | **0.165** | **0.201** | **0.108** | **0.126** | 0.077 | 0.138 | 0.105 | **0.140** | 0.094 |
| Adjusted R² | **0.186** | **0.124** | **0.162** | **0.063** | **0.086** | 0.034 | 0.070 | 0.031 | **0.086** | 0.036 |
| Residual Std. Error | **0.902 (df = 125)** | **0.936 (df = 124)** | **0.928 (df = 121)** | **0.988 (df = 119)** | **0.901 (df = 108)** | 0.927 (df = 107) | 0.998 (df = 76) | 1.036 (df = 73) | **0.918 (df = 79)** | 0.954 (df = 77) |
| F Statistic | **6.004 (df = 6; 125)** | **4.081 (df = 6; 124)** | **5.082 (df = 6; 121)** | **2.406 (df = 6; 119)** | **3.123 (df = 5; 108)** | 1.780 (df = 5; 107) | 2.029 (df = 6; 76) | 1.427 (df = 6; 73) | **2.583 (df = 5; 79)** | 1.605 (df = 5; 77) |
| F Statistic (p-value) | **<0.001** | **<0.001** | **<0.001** | **0.0313** | **0.011** | 0.123 | 0.0719 | 0.216 | **0.0324** | 0.169 |

*Note:* Results of linear regression models, not adjusted for multiple comparisons. Estimates are standardized betas; 95% Confidence Interval reported in brackets. Values in bold are statistically significant at a level $p < 0.05$ (two-sided).

*df* degrees of freedom.

Controlling for age, gender, childhood trauma, and household income at baseline, PASS-content, perceived social support, and perceived good stress recovery were negatively prospectively associated with SR at the different time scales (Table 2, Supplementary Table 8). PASS-content explained 12.5 % of the variance in SR(B0-B2). In the most stress-exposed participants (top two terciles of mean E between B0 and B2), all three factors were also predictive of SR, except for PASS-process for the short interval of 9 months after B1 (Supplementary Tables 9 and 10). Mediation analyses were well powered (Methods) and showed the expected mediation of social support effects on SR via PASS-content and of PASS-content effects on SR via good stress recovery (Fig. 4).

As an interim conclusion, observations using PASS-content to index positive appraisal style in two independent, demographically different German samples are compatible with the theoretical claims

that PAS is a resilience factor, that it mediates the effects of other resilience factors (social support), and that it acts on resilience by shaping stress responses in an optimal way (facilitating stress recovery).

**Interventional sample: RESPOND-RCT Spain**

RESPOND-RCT Spain is a multicenter, parallel-group, analyst-blinded RCT that explored the effectiveness of a stepped-care program versus enhanced care as usual (eCAU) among healthcare workers with psychological distress in the Communities of Madrid and Catalonia (Spain). The stepped-care program comprised two psychological interventions: initially, a guided stress management course based on the SH+ booklet called Doing What Matters (DWM) in Times of Stress, and potentially an individual intervention based on cognitive–behavioral therapy called Problem Management (PM + ) (see

**A**

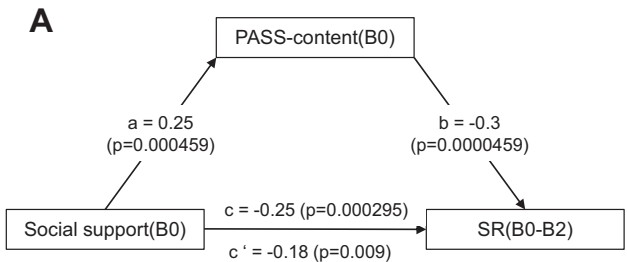

**C**

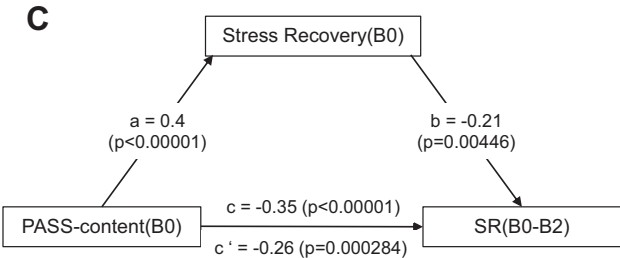

**B**

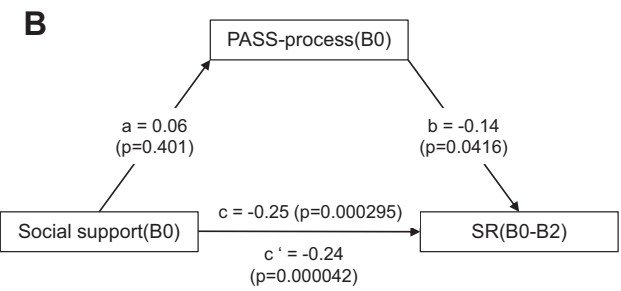

**D**

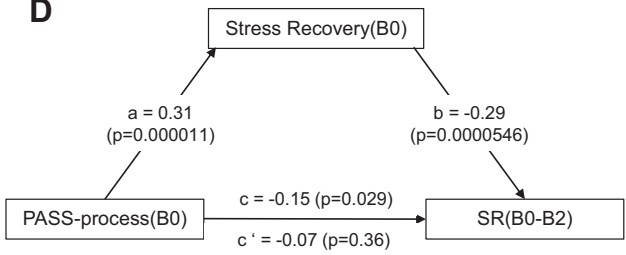

**Fig. 3 | MARP: Mediation analysis based on linear regression models[85].** The effect of perceived social support(B0) on SR(B0-B2) was mediated by PASS-content(B0) (mean bootstrapped indirect effect based on random resamples ab: 0.07, 95% CI [-0.15, -0.01]) (**A**); this effect was not significant for PASS-process (ab: -0.01, 95% CI [-0.04, 0.01]) (**B**). Perceived good stress recovery(B0) mediated the effect of PASS-content(B0) on SR(B0-B2) (ab: -0.08, 95% CI [-0.17, -0.01]) (**C**), and the effect of PASS-process(B0) (ab: -0.09, 95% CI [-0.17, 0.03]) (**D**). a, effect of predictor on mediator; b, effect of mediator on outcome; c, total effect of predictor on outcome; c', direct effect of predictor on outcome removing the mediator.

Fig. 5). Between November 2021 and March 2022, 232 participants were randomized to the stepped-care program or eCAU using a 1:1 allocation ratio (for a full RCT flow diagram see Supplementary Fig. 1). During this period, there was a new peak of hospital admissions and patients in intensive care in Spain due to COVID-19[44] and a concomitant substantial disruption of normal working conditions[40]. In the trial sample, 93% of participants were or had been directly involved in caring for COVID-19 patients, and 59% had been infected with SARS-CoV-2[40].

Average T0 PHQ-ADS scores were 20.47 (SD = 8.4). On this instrument, values above 20 indicate moderate levels of depression/anxiety[45]. One hundred and twenty-four (33%) participants scored above the recommended cut-off for depression (9 on the PHQ-9 subscale) and 134 (58%) above the cut-off for anxiety (9 on the GAD-7 subscale)[46,47]. This suggests substantial distress in the sample, in line with participants being pre-selected for a K10 value of ≥16. Groups did not differ on these characteristics (Supplementary Table 13). A detailed trial description can be found in the study protocol[36] and the publication of the primary analysis[40]. All secondary analyses reported in this paper were performed on the entire sample (intention-to-treat analyses), as specified in the study protocol[36]. In all analyses, we controlled for age, gender, and education at baseline (see Methods for details).

Over the trial, the most frequently reported life event was 'serious illness, accident, or diagnosis of disease experienced by me or a close person' (reported by an average of 67.2 participants per time point (SD = 75.3)). 'Death of a close person (e.g. family member or close friend)' was rated most burdensome (from 0 to 4: $M = 3.13$, SD = 0.8). The most frequently reported daily hassles were high demands/high workload/time pressure (reported by an average of 112.2 participants per time point (SD = 38.2)), less physical activity than usual ($M = 79.7$, SD = 30.2), and difficulty combining social life with work ($M = 74.0$, SD = 28.3). Participants were not asked to report hassle severity.

The aggregated stressor exposure score E (see Methods) decreased over time (effect of time: $B = -2.02$, 95 % CI [-2.45, -1.58], $p < 0.001$; Fig. 6A, Supplementary Table 14). The interaction between intervention group and time ($B = -0.77$, 95 % CI [-1.38, -0.15], $p = 0.015$) was significant, such that E decreased more in the intervention group. The effect of group was statistically not significant ($B = -0.23$, 95 % CI [-2.23, 1.76], $p = 0.819$). The interaction effect may be explained by effects of the intervention on stressor perception and reporting or on stressor exposure (e.g., decreased risk taking, better problem solving, or better stressor avoidance). The effect highlights the need to control for exposure differences also in randomized trials. Mental health problems P also decreased over time and more so in the intervention group ($B = -1.66$, 95 % CI [-2.29, -0.93], $p < 0.00001$; covariate-controlled) (Fig. 6B).

E explained 64.1% of variance in P across assessment time points and participants, including random effects (conditional $R^2$; marginal $R^2 = 43.2\%$, including fixed effects only). Unlike E and P, SR did not show a significant effect of time ($B = -0.037$, 95 % CI [-0.11,-0.04], $p = 0.328$; covariate-controlled) (Fig. 6c). There was a significant group by time interaction ($B = -0.15$, 95 % CI [-2.25, -0.042], $p = 0.00623$), reflecting a reduction specifically in the intervention group. The effect of group was statistically not significant ($B = 0.13$, 95 % CI [-0.18, 0.44], $p = 0.414$). These findings can be interpreted as the intervention promoting resilience against the exacerbating effects of stressor exposure on internalizing symptomatology (for effect sizes, see Supplementary Table 15a and 15b).

In congruence with MARP and LORA, baseline PASS-content (T0) scores prospectively and negatively predicted SR across T1 to T3 ($B = -0.028$, 95 % CI [-0.052, -0.006], $p = 0.015673$; covariate-controlled). There were no significant time or group effects (time: $B = -0.05$, 95 % CI [-0.16,0.06], $p = 0.402$; group: $B = 0.11$, 95 % CI [-0.71, 0.33], $p = 0.482$).

Importantly, the intervention affected PASS-content (Fig. 6d). There was a significant group by time interaction (B = 1.02, 95 % CI

**Table 2 | LORA: Prediction of stressor reactivity (SR) by PASS-content and PASS-process, controlling for baseline (B0) covariates**

| Predictor time point (battery) Outcome interval (SR score) | B0 | | | | | | B1 | | | |
|---|---|---|---|---|---|---|---|---|---|---|
| | BO-B2 (~3 yrs) | | BO-B1 (~1.5 yrs) | | 3 monitorings post B0 (~9 m) | | B1-B2 (~1.5 yrs) | | 3 monitorings post B1 (~9 m) | |
| | PASS-content | PASS-process | PASS-content | PASS-process | PASS-content | PASS-process | PASS-content | PASS-process | PASS-content | PASS-process |
| PAS | **−0.209** | **−0.110** | **−0.221** | **−0.122** | **−0.191** | **−0.117** | **0.216** | **−0.104** | **−0.234** | **−0.075** |
| | (−0.252, −0.165) | (−0.155, −0.064) | (−0.267, −0.175) | (−0.170, −0.073) | (−0.247, −0.135) | (−0.174, −0.060) | (−0.278, −0.155) | (−0.166, −0.042) | (−0.298, −0.170) | (−0.139, −0.010) |
| | $p < 0.00001$ | $p < 0.00001$ | $p < 0.00001$ | $p < 0.00001$ | $p < 0.00001$ | $p < 0.00001$ | $p < 0.00001$ | $p = 0.002$ | $p < 0.00001$ | $p = 0.025$ |
| Age | −0.001 | −0.001 | −0.003 | −0.003 | 0.003 | −0.002 | 0.002 | 0.003 | 0.001 | 0.003 |
| | (−0.007, 0.005) | (−0.007, 0.005) | (−0.010, 0.003) | (−0.010, 0.003) | (−0.010, 0.005) | (−0.010, 0.005) | (−0.006, 0.010) | (−0.005, 0.012) | (−0.007, 0.010) | (−0.006, 0.011) |
| | $p = 0.676$ | $p = 0.775$ | $p = 0.289$ | $p = 0.357$ | $p = 0.460$ | $p = 0.570$ | $p = 0.602$ | $p = 0.408$ | $p = 0.773$ | $p = 0.527$ |
| Gender | **0.307** | **0.286** | **0.332** | **0.309** | **0.297** | **0.277** | **0.244** | **0.236** | **0.235** | **0.242** |
| | (0.214, 0.399) | (0.190, 0.382) | (0.233, 0.430) | (0.207, 0.411) | (0.177, 0.416) | (0.155, 0.400) | (0.113, 0.374) | (0.103, 0.369) | (0.100, 0.371) | (0.102, 0.381) |
| | $p = 0.000$ | $p = 0.000$ | $p = 0.000$ | $p = 0.000$ | $p = 0.00001$ | $p = 0.00002$ | $p = 0.0003$ | $p = 0.001$ | $p = 0.001$ | $p = 0.001$ |
| Childhood trauma | **0.005** | **0.007** | **0.007** | **0.009** | **0.009** | **0.011** | 0.004 | 0.006 | 0.005 | **0.008** |
| | (0.001, 0.010) | (0.003, 0.012) | (0.002, 0.012) | (0.004, 0.014) | (0.003, 0.014) | (0.005, 0.016) | (−0.002, 0.010) | (−0.0002, 0.012) | (−0.001, 0.012) | (0.002, 0.015) |
| | $p = 0.020$ | $p = 0.003$ | $p = 0.004$ | $p = 0.0004$ | $p = 0.003$ | $p = 0.0004$ | $p = 0.180$ | $p = 0.059$ | $p = 0.127$ | $p = 0.017$ |
| Income | **−0.032** | **−0.037** | **−0.029** | **−0.034** | **−0.036** | **−0.037** | **−0.049** | **−0.060** | **−0.061** | **−0.072** |
| | (−0.057, −0.008) | (−0.062, −0.011) | (−0.055, −0.003) | (−0.061, −0.006) | (−0.068, −0.004) | (−0.070, −0.005) | (−0.084, −0.013) | (−0.096, −0.024) | (−0.098, −0.024) | (−0.110, −0.034) |
| | $p = 0.011$ | $p = 0.005$ | $p = 0.029$ | $p = 0.016$ | $p = 0.028$ | $p = 0.024$ | $p = 0.008$ | $p = 0.002$ | $p = 0.002$ | $p = 0.0003$ |
| Constant | **−0.510** | **−0.534** | **−0.569** | **−0.593** | **−0.543** | **−0.588** | **−0.405** | **−0.436** | −0.343 | **−0.458** |
| | (−0.779, −0.241) | (−0.813, −0.255) | (−0.855, −0.283) | (−0.890, −0.296) | (−0.889, −0.198) | (−0.941, −0.234) | (−0.788, −0.023) | (−0.826, −0.047) | (−0.748, 0.062) | (−0.874, −0.042) |
| | $p = 0.0003$ | $p = 0.0002$ | $p = 0.0002$ | $p = 0.0001$ | $p = 0.003$ | $p = 0.002$ | $p = 0.039$ | $p = 0.029$ | $p = 0.098$ | $p = 0.032$ |
| Observations (n) | 1,034 | 1,033 | 1,034 | 1,033 | 929 | 930 | 754 | 761 | 705 | 709 |
| $R^2$ | 0.130 | 0.073 | 0.132 | 0.077 | 0.089 | 0.060 | 0.095 | 0.050 | 0.111 | 0.054 |
| Adjusted $R^2$ | 0.125 | 0.068 | 0.128 | 0.073 | 0.085 | 0.055 | 0.088 | 0.044 | 0.105 | 0.047 |
| Residual Std. Error | 0.715 (df = 1028) | 0.743 (df = 1027) | 0.762 (df = 1028) | 0.790 (df = 1027) | 0.872 (df = 923) | 0.892 (df = 924) | 0.855 (df = 748) | 0.876 (df = 755) | 0.857 (df = 699) | 0.883 (df = 703) |
| F Statistic | 30.622 (df = 5; 1028) | 16.063 (df = 5; 1027) | 31.364 (df = 5; 1028) | 17.225 (df = 5; 1027) | 18.132 (df = 5; 923) | 11.716 (df = 5; 924) | 15.621 (df = 5; 748) | 7.980 (df = 5; 755) | 17.435 (df = 5; 699) | 8.056 (df = 5; 703) |
| F Statistic (p-value) | <0.001 | <0.001 | <0.001 | <0.001 | <0.001 | <0.001 | <0.001 | <0.001 | <0.001 | <0.001 |

Note: Results of linear regression models, not adjusted for multiple comparisons. Estimates are standardized betas; 95% Confidence Interval reported in brackets. Values in bold are statistically significant at a level $p < 0.05$ (two-sided). Results for PASS-process are shown for descriptive purposes only.
*df* degrees of freedom.

[0.53, 1.49], $p < 0.0001$; covariate-controlled), such that PASS-content scores increased more over time in the intervention group. The effect of group was statistically not significant (B = -0.89, 95% CI [-2.66, 0.88], $p = 0.330$; and time: $B = 0.33$, 95% CI [-0.004, 0.66], p = 0.0534).

These results raise the possibility that intervention-induced increases in PAS mediate the beneficial intervention effects on SR. Post-hoc power analysis (Methods) showed a power of >0.9 for a sample >220, indicating that the study was sufficiently powered to detect mediation.

We therefore conducted a planned prospective mediation analysis, estimating the effect that the intervention has on the mediator (PASS-content at post-intervention, T2) and the subsequent outcome (SR at follow-up, T3; cf. Fig. 5). Analysis used the VanderWeele approach and controlled for baseline (T0) age, gender, education, PASS-content, and SR.

The total effect (te) of the intervention on SR at T3 was -5.37 (95% CI [-10.18, -0.28], $p = 0.042$), and the total natural direct effect (nde) of the intervention on SR at T3 not mediated through PASS-content at T2 was -3.46 (95% CI [-8.30, 2.38]). This was not significant ($p = 0.280$). The natural indirect effect (nie) was -2.31 (95% CI [-5.24, -0.85], $p = 0.036$), indicating that the effect of the intervention on SR at T3 was mediated through PASS-content at T2. The proportion mediated (pm) by PASS-content was 47% (95% CI [-1.35, 2.23]). When only analyzing the complete cases ($n = 135$; 63 in the intervention group and 72 in the control group), te, nie, and pm were statistically significant, but nde was not statistically significant (te = -7.49, 95% CI [-12.98, -1.93], $p = 0.004$; nde = -5.07, 95% CI [-10.74, 0.66], $p = 0.096$; nie = -2.42, 95% CI [-5.37, -0.038], $p = 0.046$; pm = 0.32, 95% CI [0.001, 1.18]). These results suggest that PAS is partially responsible for the positive effect of the multi-component intervention on resilience.

## Discussion

The Positive Appraisal Style Theory of Resilience (PASTOR) relies on the assumption that stress results from the appraisal of a stimulus or situation as threatening one's goals or needs[12]. PASTOR further assumes that individuals exhibit style-like (relatively stable, but not

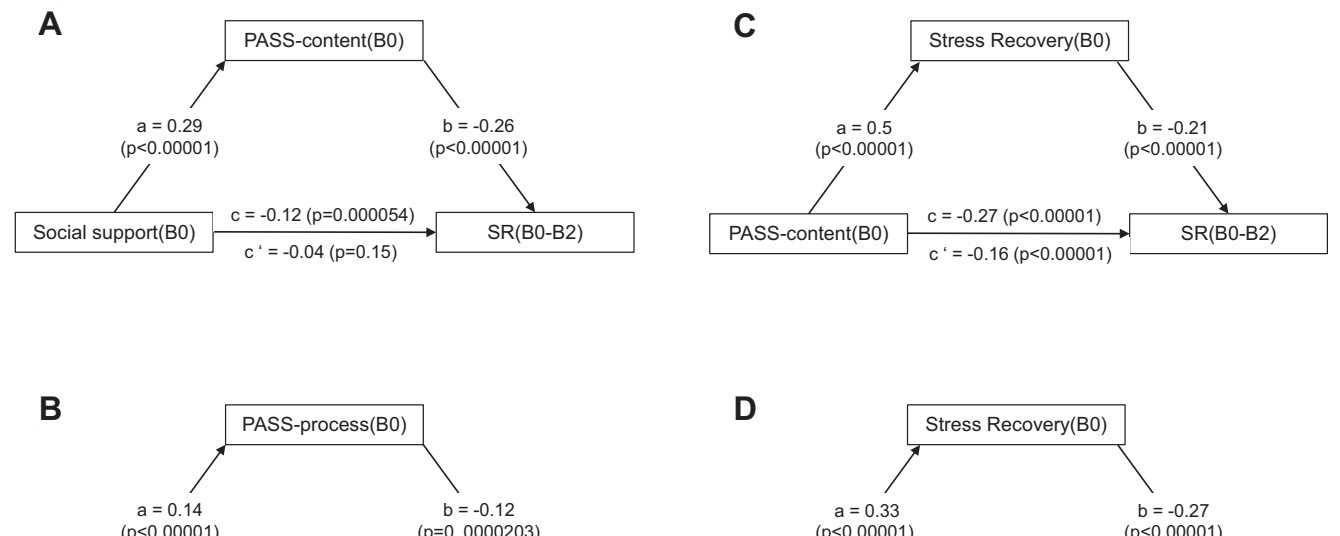

Fig. 4 | LORA: Mediation analysis based on linear regression models[85]. The effect of perceived social support(B0) on SR(B0-B2) was mediated by PASS-content(B0) (mean bootstrapped indirect effect based on random resamples ab: -0.07, 95% CI [-0.1,-0.05]) (**A**). For descriptive purposes, mediation by PASS-process(B0) (ab: -0.02, 95% CI [-0.03, -0.01]) is shown in (**B**). Perceived good stress recovery(B0) was mediated the effect of PASS-content(B0) (ab: -0.1, 95% CI [-0.14, -0.07]) on SR(B0-B2) (**C**). For descriptive purposes, mediation of the effect of PASS-process(B0) (ab: -0.09, 95% CI [-0.12, -0.06]) is shown in (**D**). a, effect of predictor on mediator; b, effect of mediator on outcome; c, total effect of predictor on outcome; c', direct effect of predictor on outcome removing the mediator. All significance levels are two-sided. No adjustments for multiple comparisons.

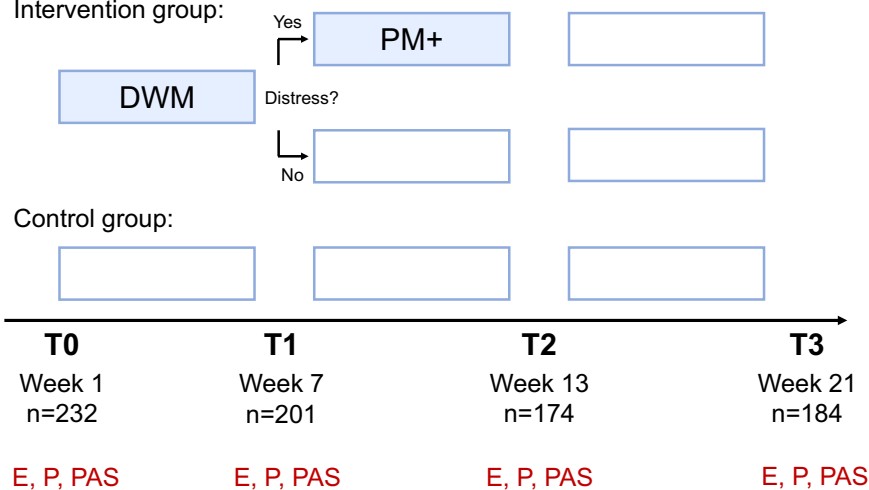

Fig. 5 | RESPOND-RCT Spain: Design. The trial sample (*N* = 232) was predominantly female (*n* = 200, 86%), with an average 37.5 years of age (SD = 10.3) at baseline. Most participants had a university degree (*n* = 192, 82%) and were nurses (*n* = 130, 56%), physicians (*n* = 50, 22%), or nursing technicians (*n* = 29, 13%). The intervention group (top panel, *n* = 115) took part in a stepped-care program consisting of Doing What Matters (DWM) and, if distress continued to be present (score ≥16 on the Kessler Psychological Distress Scale (K10)[86] five to 7 days after DWM), in Problem Management + (PM + ; *n* = 84 or 75% of participants in the intervention group). The control group (*n* = 117) received enhanced care as usual in the form of psychological first aid. Empty boxes illustrate no intervention. Stressor exposure (E), mental health problems (P), and positive appraisal style (PAS) were assessed in all participants at four time points (T0 to T3). Stressor assessment used instruments adapted to the specific population and context in prior qualitative work[36,78], one list featuring three major life events and one list featuring six general, five pandemic-related, and four population-specific stressors (Supplementary Tables 11 and 12). Mental health assessment used the Patient Health Questionnaire-Anxiety and Depression Scale (PHQ-ADS)49, a composite measure of anxiety (GAD-7) and depression (PHQ-9) symptoms. PAS assessment in the RESPOND trial was restricted to the PASS-content instrument, which had shown significant SR associations in MARP and LORA. This scale also has the advantage that it directly targets the element in appraisal (appraisal contents, or outcomes) that is hypothesized to eventually determine stress responses, rather than antecedent cognitive processes leading to these contents (as in PASS-process)[12]. Also, social support and stress recovery were not assessed. Figure adjusted from ref. 41.

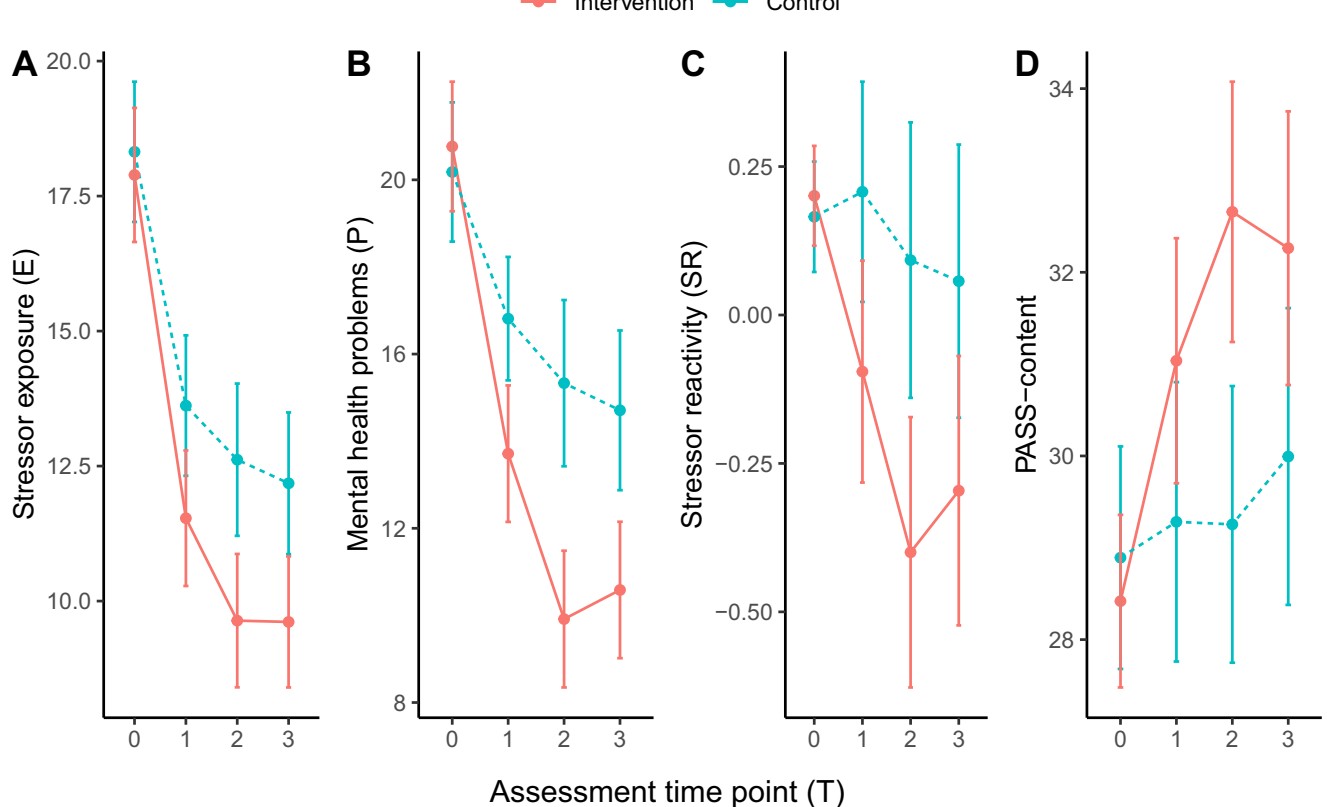

**Fig. 6 | RESPOND-RCT Spain: Results.** Estimated marginal means and 95% confidence intervals (error bars) for stressor exposure (**E**) scores (T0: $n = 104$ (Control group)/105 (Intervention group), T1: $n = 88/89$, T2: $n = 75/78$, T3: $n = 80/78$) (**A**), mental health problem (**P**) scores from the PHQ-ADS scale (T0: $n = 117/115$, T1: $n = 102/99$, T2: $n = 88/86$, T3: $n = 94/90$) (**B**), stressor reactivity (SR) scores (T0: $n = 104/105$, T1: $n = 88/89$, T2: $n = 75/78$, T3: $n = 80/68$) (**C**), and PASS-content scores (T0: $n = 114/114$, T1: $n = 98/95$, T2: $n = 84/86$, T3: $n = 92/87$) (**D**) in the intervention (red) and control (blue) groups. The effect of the intervention on mental health P has also been published elsewhere[40].

entirely temporally invariant) individual differences in the way they appraise potential threats. Individuals with a positive appraisal style (PAS) typically avoid overestimating threat magnitude/cost (catastrophizing) and threat probability (pessimism) and underestimating their coping potential (helplessness). They also avoid very unrealistically positive estimates, which would equate to trivialization, blind optimism, and delusional over-confidence, respectively. Hence, their usual appraisals on key threat appraisal dimensions range from realistic to mildly positive values (compare Fig. 1). As a result, these individuals are still able to produce stress responses as necessary to cope with challenges but also do not normally over-react. Hence, they are less likely to unnecessarily consume resources and experience exhaustion, allostatic load effects, and eventually stress-related mental health problems. Their mild tendency to under-react prevents life in a continuous alarm mode. Instead, positive appraisers find it easier to rebuild resources and to expand their behavioral repertoire and life possibilities by learning from the encounter and exploration of new situations, which negative appraisers would avoid[12].

Testing PASTOR requires valid measurement of PAS. One important consideration in the development of PAS measurement instruments is that appraisals may be generated via a heterogeneous set of cognitive processes, from unconscious, non-verbal, fast, and implicit to conscious, verbal, more slowly, and explicit[12] (for a more detailed discussion, see Supplementary Note 2). The Perceived Positive Appraisal Style Scale, process-focused (PASS-process), which we first created during the COVID-19 pandemic to serve as a tool for large-scale online surveys, aims at indexing conscious positive appraisal and reappraisal processes (e.g., 'I think that I can become a stronger person as a result of what has happened.'; 'I tell myself that there are worse

things in life.')[15]. By its very nature, it cannot index unconscious processes. By contrast, the more recently validated Perceived Positive Appraisal Style Scale, content-focused (PASS-content) focuses on the generated appraisals, that is, on how individuals typically think of stressful situations (e.g., 'I tend to see things rather optimistically.'; 'I think that I somehow always manage to get what I need.')[19]. The instrument has the advantage that it directly targets the assumed causal determinants of stress responses, i.e., the appraisals. Like PASS-process, it cannot inform about possible unconscious appraisals, which, too, may affect stress responding. This reliance on measures that are restricted to consciously accessible mental contents is a limitation of the present work. A further limitation is the known vulnerability of self-report instruments to reporting and memory biases[19]. In particular, healthy humans tend to uphold a stable and positive self-model and also to search for stability in their world-model, and this source of bias is likely to shape the way they think and communicate about internal or external threats to these important goals. This may have the ironic consequence that individuals who are more successful in protecting their self- and world-models by recurring to partly illusionary, positively biased stressor appraisals are probably also mentally healthier[19,48,49]. In this sense, it is also possible that the methodological biases inherent to our PAS instruments may not be a source of error but rather provide some key information on an important source of mental health. This may apply in particular to the PASS-content scale, high scores on which can be interpreted as reflecting a positive self-/world-perception[19].

Our results show that PAS—measured with PASS-content—prospectively predicts low stressor reactivity over long time frames of >3 years. PASS-process did not show significant association in the

smaller exploratory sample MARP, but did predict SR in the larger sample LORA. These findings provide important insight, since the initial studies (using PASS-process) had either tested association with SR scores derived from a single, concurrent time point (asking participants about their stressor exposure and changes in mental health in the past weeks[15,16]), or from short prospective intervals of 5 weeks[17] or 6 months[18]. Hence, we can now say that PAS is a resilience factor with long-term predictive value, at least if assessed with PASS-content.

Another insight from the available data is that PAS is a resilience factor across different populations: European adults[15,17], mental health practitioners from various countries[16], Dutch Parkinson patients[18], German young adults (MARP), German adults (LORA), and Spanish health care workers (RESPOND-RCT Spain). (We also note that PASS-process was inversely related to internalizing symptoms and to the strength of symptom network connections in Dutch elderly persons during the pandemic[50].) This conclusion is still limited by the absence of data from non-European and from non-adult samples.

A final insight is that PAS appears to protect against various types of stressors. Thus, of the five studies focusing on individuals confronted with the pandemic[15–18,36], two (the mental health practitioner study[16] and RESPOND-RCT Spain) used samples where the pandemic-related challenges stemmed at least in part from professional demands. The MARP and LORA assessments, on the other hand, were largely performed before the pandemic, and especially the shorter early SR time frames (B0-B1, first 9 months post B0) were not or minimally influenced by this large-scale stressor. Our detailed analyses of the stressor exposure in these samples further substantiate that the participants experienced qualitatively different exposure.

In sum, the available results suggest that PAS may be a universal resilience factor valid in different types of population and in different stressor situations. One important question for future work is whether PAS also exerts protective effects in populations exposed to more extreme stressors than present in the samples analyzed so far, including severe potentially traumatic events.

What the existing data cannot answer is whether the protective effects of PAS extend to different types of mental health problems, notably to symptoms beyond the internalizing spectrum (i.e., mainly anxiety and depression). An analysis of mental health problems specifically observed in mental health practitioners, featuring aspects of burnout and secondary traumatic stress, suggested PAS may be less protective against these impairments[16]. Protection against externalizing or psychotic symptoms has not been tested yet.

Our present data also showed that PAS is a relatively stable individual-differences factor, with ICCs above 0.7 in the case of our now preferred instrument, PASS-content. This raised the question of whether PAS can change, as the original conceptualization of PAS as having plasticity and being modifiable by life experiences or interventions suggests[12]. We tried to answer this question alongside the question of whether PAS is a proximal resilience factor that integrates the effects of various other factors on resilience. The finding that PAS mediates effects of social support, now consistently observed in the four initial studies[15–18] as well as here in MARP and LORA, aligns with the idea in PASTOR that different kinds of beneficial life circumstances or experiences (as well as protective predispositions, skills, or behavioral styles) all promote resilience because they eventually bias stressor appraisal towards more positive values. This led us to ask whether a broad psychosocial-behavioral intervention, combining various elements from traditional and more recent therapy approaches[36], would enhance PASS-content scores and whether this would in turn mediate the expected beneficial effects of the intervention on resilience. This was confirmed in the RESPOND-RCT Spain study. Both steps of the intervention (DWM and PM + ), though targeting different constructs, appeared to have comparable effects on PAS (compare Fig. 6c, d), further nurturing the idea that PAS is an integrative, proximal resilience factor that can be indirectly enhanced via different routes.

Altogether, our findings suggest that PAS is clearly stable, but also malleable.

The randomized controlled design of that study and the clear hypothesis-driven nature of our analysis further allow us to forward the hypothesis that the influence of PAS on resilience was causal. This possible causal effect may well have been exerted by PAS optimizing stress reactions, as is indirectly indicated by the mediation of PAS effects on resilience via good self-perceived recovery from stressors, observed in the initial studies[15–17] as well as now in MARP and LORA. We, however, emphasize that the intervention targeted a wide range of potential factors, which included appraisal but were not limited to this construct. We report largely unbiased estimates of causal effects of PAS on resilience. However, these estimates are still hindered by unmeasured confounding, reverse causation, or indirect manipulation of the exposure. Future intervention studies that directly manipulate PAS will help make stronger causal claims that strengthen theory and inform intervention development. These studies should focus on the PASS-content instrument as an intervention target and effect mediator, which in the present work is more reliable and more strongly associated with SR than PASS-process and also has higher construct validity (see above).

Overall, our findings paint a coherent mechanistic picture of resilience: factors that promote a tendency to appraise stressors in a mildly unrealistically positive fashion lead to optimally regulated (fine-tuned, situation-appropriate) stress responses. This, in turn, preserves bodily and cognitive capacities that are crucial for long-term mental functioning. One major challenge for future work will be to complement the existing self-report instruments for PAS with more objective, task-based measures that implicitly assess biases via behavioral or physiological responses to threat-related stimuli. Such biases may become apparent in particular in reactions to stimuli that are ambiguous with respect to their threat value. A suitable task should manipulate threat values on PASTOR's three key threat appraisal dimensions of cost/magnitude, probability, and coping potential. A further challenge will be to establish the generalizability of our observations to a broader range of populations and types of adversity to thus definitively establish PAS as a universal resilience factor. Yet another challenge will be to empirically compare and integrate PASTOR with theories emphasizing the role of flexible coping and emotion regulation for resilience (e.g. refs. 51–56,). Flexibility-based theories assume that coping or emotion regulation strategies are not by themselves adaptive or maladaptive. Rather, the adaptive value of a regulatory strategy is determined by the situation or context in which the strategy is used. Therefore, it is important to be able to flexibly shift between strategies as a function of situational demands. From the perspective of PASTOR, a tendency for positive appraisal makes it less likely that the person is overwhelmed by a stressful situation, which in turn makes it easier for them to employ different coping strategies in a situation-adapted fashion. Thus, positive appraisal permits regulatory flexibility, and regulatory flexibility can be an aspect of optimized stress response regulation. However, PASTOR does not claim that regulatory flexibility is always and necessarily an element of resilient responding to stressors[12,14].

Irrespective of further theoretical progress, our results highlight promising avenues for promoting resilience. Next to broad and multifaceted interventions that also target more distal resilience factors, like the program employed in our RCT, resilience might also be enhanced by interventions specifically targeting PAS as a key proximal factor. This may involve positive mindset interventions[57], social-psychological interventions[58], or dedicated positive reappraisal[59,60] or bias modification[61] trainings. Another promising approach might be to simultaneously play on a more distal factor that is mediated in its effect by PAS, such as social support, and on PAS. Finally, it could be interesting to explore the differential contributions of the DWM and PM+ components of our RCT to PAS improvements, to thus identify the

most efficient approach for shorter interventions and to better understand the relationships of their constitutive elements with appraisal style. Such mechanistically more specific resilience trainings may become an important tool in the global fight against stress-related disorders.

# Methods

All analyses were conducted using R (4.2.3)[62] and R studio (Version 2023.090)[63]. The following packages were used: cmaverse (0.1.0)[64], tidyverse (2.0.0)[65], psych (2.3.3)[66], lme4 (1.1–32)[67]. Code is available on: https://doi.org/10.17605/OSF.IO/D4EXM.

## Observational discovery sample: MARP

**Design.** MARP is a multi-modal longitudinal observational study being conducted by the University Medical Center of Johannes Gutenberg University and the Leibniz Institute for Resilience Research in Mainz, Germany. At baseline (B0), MARP has included $N = 200$ participants without any psychiatric diagnosis, aged between 18 and 21 years. Participants were selected based on previous experience of stressful life events (a minimum of 3 before the age of 18). This criterion was chosen to reflect the average number of past life events to be expected in the study population based on population-based data from the same geographical region. A population-average number of life events was considered a risk factor, extrapolating from other work[68]. We thereby intended to enrich the sample for risk for mental health problems. Exclusion criteria included current psychological or neurological disorder, taking psychoactive medication, and physical illness that affect mental health. Sampling was non-representative. Inclusion for B0 took place between July 2016 and March 2019. Participants gave their written and informed consent. Ethical approval was granted by the Medical Board of Rhineland-Palatinate, Mainz, Germany. Participants received reimbursement for their efforts.

The MARP design involves regular testing batteries (B0, B1, B2, …), planned to be administered -1.75 years apart, as well as online assessments planned every 3 months (T1, T2, T3, …) (Fig. 2). The first online monitoring (T0) only serves to acquaint participants with the procedure and is not analyzed. The batteries consist of online questionnaires (in German) covering sociodemographic information, mental health measures, as well as lifestyle, psychosocial, and psychological constructs. On-site testing, such as neuroimaging as well as biospecimen collection, is also conducted at each battery but is not the subject of this study. The 3 monthly online assessments serve to regularly monitor mental health problems and stressor exposure (see Measures). Questionnaire data was collected with secuTrial© (an electronic data capture system, www.secutrial.com). For an overview and first methodological publications, see refs. 20,21.

**Data cleaning and preparation.** A data freeze was performed November 11th 2022. In this data set, in some cases and particularly during the pandemic, the interval between battery administrations could be more or less than 21 months. If the interval between two battery administrations exceeded 21 months, more online monitorings than planned could take place. Also, the online monitorings planned concordant with the batteries (T7/B1, T14/B2, cf. Fig. 2) were conducted separately from the corresponding battery assessments and could therefore take place before or after a battery. Finally, participants were allowed to miss online monitorings. As a result, the number of online monitorings between battery administrations varied between participants. No data was imputed. Of the 200 participants at inclusion, 132 participants could be included in the longitudinal analysis, since they provided the minimally accepted number of four completed online monitorings between B0 and B2. Their demographics did not significantly differ ($p < 0.05$) from the full baseline sample on any of the variables given in Supplementary Table 1 (for further details, see Supplementary Note 3).

## Measures

**Positive appraisal style (PAS).** PAS is assessed in each battery (B0, B1, …) with two recently developed scales that were validated in the MARP and LORA samples (reported in ref. 19). Firstly, the 14-item Perceived Positive Appraisal Style Scale, content-focused (PASS-content) (internal reliability: Cronbach's $\alpha = 0.87$, McDonald's $\omega = 0.88$) measures the self-reported frequency with which someone produces thoughts that amount of positive appraisal, such as "I think that every difficult situation will end eventually", in stressful situations. Answers are scored on a 4-point Likert scale from 1 ("never") to 4 ("almost always"). A higher sum score (range 14 to 56) denotes more frequent positive appraisal thoughts, that is, a more positive appraisal tendency. The internal reliability in this sample is good ($\alpha = 0.86$, $\omega = 0.86$). Secondly, the 10-item Perceived Positive Appraisal Style Scale, process-focused (PASS-process) ($\alpha = 0.78$, $\omega = 0.85$) assesses the self-reported frequency of mental operations (cognitive strategies and tactics) that someone employs in stressful situations and that can generate positive appraisal contents. These cognitive processes include, for instance, acceptance ("I think that I have to accept the situation"), positive reframing ("I think that I can become a stronger person as a result of what has happened"), or distancing ("I try to look at the situation from an objective perspective"). Participants rate their use on a 5-point Likert scale from 1 ("(almost) never") to 5 ("(almost) always"). A higher sum score (range 10 to 50) signifies a more frequent use of such positive appraisal and reappraisal processes[19]. In the MARP sample, the internal reliability of PASS-process is $\alpha = 0.72$, $\omega = 0.73$.

**Perceived social support.** The 14-item version of the perceived social support questionnaire by Fydrich et al.[69] ($\alpha = 0.94$) is included in all batteries. Participants rate their agreement on a 5-point Likert scale from 1 to 5. A higher mean score denotes higher perceived social support.

**Perceived good stress recovery.** Perceived good stress recovery is assessed using the 6-item Brief Resilience Scale[70] ($\alpha = 0.8$ to 0.91) in each battery. Respondents rate their agreement on a 5-point Likert scale from 1 to 5. A higher mean score denotes a higher self-perceived ability to bounce back.

**Mental health problems (P).** Mental health problems are captured at every assessment (batteries and online monitorings) with the 28-item General Health Questionnaire (GHQ-28)[41] ($\alpha = 0.9$ to 0.95)[42], covering general internalizing mental health symptoms, including depressive and anxious symptoms. Participants assess their symptoms over the previous weeks, rating each item on a scale from 0 to 3. All items are then summed into a total score (range 0 to 84). Only the online monitoring data were analyzed here.

**Stressor exposure (E).** Stressors are assessed at every online monitoring (T0, T1, T2, …). This includes a list of 27 life events (macro-stressors) adapted from Canli's Life Experience Questionnaire[71]. Participants report which life events occurred over the past 3 months and, if it occurred, rate its severity (5-point Likert scale, from 1 ("not at all burdensome") to 5 ("very burdensome")). To quantify life event exposure independent from severity, a sum count of the reported life events was calculated at each time point, as previously described[23]. Next to major events, the accumulation of daily hassles (daily stressors, microstressors) can also have a strong negative impact on mental health[72,73]. Daily hassles are assessed with the Mainz Inventory of Microstressor[74], a comprehensive list of 58 commonly occurring microstressors. Each item is rated in two ways. First respondents indicate how many days out of the past seven a stressor occurred. Second, participants rate the severity of a reported stressor on a 5-point Likert scale. To quantify daily hassles exposure independent from severity, a sum count of the number of days was calculated across

all reported daily hassles at each time point[23]. A total E score aggregating exposure to life events and daily hassles was then computed at each time point as the mean of the z-scored life events and daily hassles counts, following a predefined procedure[23].

**Stressor reactivity (SR) score.** The sample's normative stressor reactivity, that is the E-P relationship, was determined by regressing participants' average P scores of the first 9 months (covering the first three online monitorings T1 to T3) onto their average E score of the same period, as predefined for all analyses of the data set while the study is still ongoing[23]. The relationship was linear and was not improved by adding a quadratic term ($\chi = 0.89$ (df = 1), $p = 0.3373$). For each outcome interval of interest (e.g., B0 to B1; cf. Table 1), the average E and P scores in that interval were then used to calculate individual SR scores as the residuals to the normative E-P line. The required minimum numbers of completed online monitorings per outcome interval were four for B0-B2, three for both B0-B1 and B1-B2, and two for the 9 months after a battery. A negative SR score indicates that this individual displayed less mental health symptoms than others with similar stressor exposure, whereas a positive SR score indicates a higher reactivity than the sample[23]. To assess the variance of P explained by E across all time points, a separate linear mixed model with random slope and intercept was fitted across all time points and participants. Variance explained ($R^2$) was computed following the method for mixed models outlined by refs. 75,76.

**Analyses**

**Prediction analyses..** For SR prediction (Table 1 and Supplementary Tables 3–5), separate multivariate regression analyses were calculated per predictor and outcome interval. Covariate selection followed the technical procedures and criteria used in the initial studies[15–17], applied to the present data: age and gender at baseline (B0) were always included; further baseline covariates were selected based on $p$-values below 0.2 in univariate regression models on SR(B0-B2) (see Supplementary Table 16). The only deviation from this decision-making procedure was that we also included the number of online monitorings in the B0-B2 interval for analyses with SR(B0-B2), given participants could deviate strongly in this variable (see Table 3).

**Mediation analyses..** Prospective mediation analyses were based on linear regression models using the *mediate* function of the R package *psych*[66]. The indirect effect was estimated as a mean bootstrapped effect based on random resamples with bootstrapped confidence intervals. Power analyses following Schoemann et al.[77] indicated all mediation analyses were underpowered (see Table 4).

**Observational replication sample: LORA**

**Design.** LORA (LOngitudinal Resilience Assessment) is a multi-modal longitudinal observational study being conducted by the University Medical Center of Johannes Gutenberg University Mainz, Germany, and the University Hospital of Goethe University in Frankfurt am Main, Germany, since 2017. At baseline (B0), LORA included participants without any psychiatric diagnosis from the Rhine-Main area in Germany. Inclusion criteria included: being between 18 and 50 years old at study entry, proficiency in German, no lifetime diagnoses of

chronic mental disorders such as schizophrenia or bipolar disorder, no organic mental disorders, substance dependence syndromes (other than nicotine), and no other current severe axis I disorder or current severe medical conditions. Inclusion took place between October 2016 and July 2019 via convenience sampling and was not representative. Participants gave written informed consent. Ethical approval was granted by the Medical Board of Rhineland-Palatinate, Mainz, and the Ethics Committee of the Department of Medicine at the Goethe University Frankfurt. Participants received reimbursement for their efforts.

The LORA design involves regular testing batteries (B0, B1, B2,…), planned ~1.5 years apart, as well as online assessments planned every 3 months (T1, T2, T3, …) (cf. Fig. 2). The first online monitoring (T0) only serves to acquaint participants with the procedure and is not analyzed. The batteries consist of online questionnaires (in German) covering sociodemographic information, mental health measures, as well as lifestyle, psychosocial, and psychological constructs. On-site testing, such as behavioral testing as well as biospecimen collection, is also conducted at each battery but is not the subject of this study. The 3-monthly online assessments serve to regularly monitor mental health problems and stressor exposure (see Measures). As with MARP, questionnaires were collected with secuTrial© (an electronic data capture system, www.secutrial.com). For a study protocol and detailed sample characterization, see ref. 74.

**Data cleaning and preparation.** Data collected up to April 2022 were included in these analyses. Unlike in MARP, the design of the study did not allow for individual variation in the number of scheduled online monitorings between battery administrations. Therefore, all participants had a possible maximum of five online monitorings between B0 and B1 and between B1 and B2. Only participants providing the minimum number of completed online montorings, defined as in MARP, were analyzed (1034 out of the 1191 included participants). Their demographics did not significantly differ ($p < 0.05$) from the full baseline sample on any of the variables given in Supplementary Table 6.

**Measures and analyses.** All measures were as in MARP, with the exception that the severity of daily hassles and life events were rated from 0 to 4. The internal reliability of the PAS measures were similar to MARP for PASS-content ($\alpha = 0.86$, $\omega = 0.86$), but slightly lower for PASS-process ($\alpha = 0.56$, $\omega = 0.57$). All analyses were analogous, with the exception that the number of online monitorings was not included as a covariate (see Supplementary Table 17 for covariate selection). For the mediation analyses, we concluded that they were sufficiently powered, based on the power calculations in MARP (Table 4).

**Interventional sample: RESPOND-RCT spain**

**Design.** RESPOND-RCT Spain is a multicenter, parallel-group, analyst-blinded RCT that examines the effectiveness of a stepped-care program of Internet-based behavioral interventions versus enhanced care as usual (eCAU) among healthcare workers with psychological distress. Recruitment of participants lasted from November 3, 2021, to March 31, 2022. Data collection for the primary endpoint concluded on August 21, 2022.

**Table 3 | MARP: Descriptive statistics of number of online monitorings (starting with T1) and time intervals between battery administrations in the included sample (N = 132)**

| | Number of online monitorings | | | | | | Number of months | | | | |
|---|---|---|---|---|---|---|---|---|---|---|---|
| | **Mean** | **SD** | **Median** | **Min** | **Max** | **Mean** | **SD** | **Median** | **Min** | **Max** |
| **B0 to B1** | 7.2 | 1.4 | 7 | 1 | 9 | 23.5 | 2.0 | 23.6 | 18.9 | 30.6 |
| **B1 to B2** | 6.8 | 1.8 | 7 | 1 | 11 | 20.6 | 2.8 | 20.5 | 13.4 | 29.6 |
| **B0 to B2** | 13.6 | 3.4 | 14 | 2 | 18 | 43.0 | 2.6 | 44.2 | 36.0 | 51.3 |

**Table 4 | MARP: Power calculations for mediation analyses. Monte Carlo power analysis for an indirect effect >0 by Schoemann et al.[77] (1000 repetitions, 20000 Monte Carlo draws per repetition)**

| X | M | Y | *N* for power > 0.8 | Power at 132 (sample size) |
|---|---|---|---|---|
| Social support | PASS-content | SR(B0-B2) | 190 | 0.79 |
| Social support | PASS-process | SR(B0-B2) | 645 | 0.61 |
| PASS-content | Stress recovery | SR(B0-B2) | 325 | 0.15 |
| PASS-process | Stress recovery | SR(B0-B2) | 190 | 0.85 |

The target population of the original study were healthcare workers who reported psychological distress. Inclusion criteria were: being employed by the Department of Health in Madrid or Catalunya, scoring ≥16 on the Kessler Psychological Distress Scale (K10), and literacy in Spanish or Catalan. Persons who would require immediate hospitalization, had a severe mental disorder, severe cognitive impairment, were at risk of suicide or harm of self or others, and those who initiated or changed pharmacotherapy or psychological treatment in the past 8 weeks were excluded from participation. There were no withdrawal criteria.

The stepped-care intervention consisted of Doing What Matters (DWM) and Problem Management Plus (PM + ). Each intervention was delivered over 5–6 weeks. DWM is a guided self-help intervention. Material was made available online, and a helper would provide phone-based or message-based support on a weekly basis. Participants who scored ≥16 on the K10 5 to 7 days after DWM would step up to PM + . PM+ consisted of online group sessions of 60 min. The control group received enhanced care as usual (eCAU), which consisted of psychological first aid[36]. Outcomes, covariates, and the mediator were measured at baseline (T0) and three assessment points (T1 to T3) (see Fig. 5).

Participants provided written informed consent and did not receive any compensation for their participation, in compliance with the research protocols approved by the Ethics Committees at Hospital Universitario La Paz in Madrid (identifier: PI 4857) and Parc Sanitari Sant Joan de Déu in Barcelona (identifier: PIC 129-21). The trial protocol was prospectively registered on ClinicalTrials.gov (identifier: NCT 04980326). The full study protocol[36] and the results of the primary analysis (outcome evaluation)[40] are available elsewhere.

### Measures

**Positive appraisal style (PAS)..** PAS was assessed using a preliminary version of the Perceived Positive Appraisal Scale Style, content-focused (PASS-content)[19]. The published version has 14 items, whereas the version used in this trial has 12 items and a sum score range from 12 to 48. The items "I tend to see things rather optimistically" (Item 6) and "I think life is wonderful after all" (Item 9) are missing in this version. The correlation between the 14 time and the 12-item versions in LORA is 0.99, indicating sufficient overlap. The internal reliability is similar to that of the full measure (α=0.82, ω = 0.82).

**Mental health problems (P)..** The 16-item Patient Health Questionnaire-Anxiety and Depression Scale (PHQ-ADS)[45] was used to assess mental health. The sum score ranges from 0 to 48.

**Stressor exposure (E)..** Stressor exposure was measured by a stressor list that has been adapted for this sample, as described in preparatory qualitative work[78] and the study protocol[36]. Briefly, life events were adapted from the list of life events used in the first study on PAS during the COVID-19 pandemic[15] (see Supplementary Table 12). Daily hassles were based on the Mainz Inventory of Microstressors[74], used in MARP and LORA, and the stressor list used in above COVID-19 study[15]. The most frequently reported items in the LORA sample and the COVID-19 study were considered for inclusion and judged based on their relevance to the sample. Additionally, new items were created based on

qualitative interviews with healthcare workers who identified relevant and frequent stressors specific to the sample. The final list included three life events, five COVID-19-related daily hassles, six general daily hassles, and four population-specific daily hassles (see Supplementary Table 13). Life events that occurred in the past 2 months or since the last assessment were rated from 0 to 4, indicating the severity of this event from "This situation did not happen" to "Severe impact". The life events were dichotomized to indicate whether they occurred or not (maximum sum score: 3). The frequency of the 15 daily hassles over the past 2 weeks was rated on a Likert-scale from 0 to 3 ("did not happen/almost never", "sometimes", "often", "(nearly) every day"), such that a maximum sum score would be 45. A total E score was computed as the sum of dichotomized life events and daily hassles at each time point. This way of scoring stressor exposure explained more variance in P (see below) than averaging the z-scores of life events and daily hassles (E scoring procedure for MARP and LORA). Participants reported a mean stressor load E of 13.50 (SD = 7.02) out of a possible 48 over all time points. (See Supplementary Table S12 for details on stressor exposure.)

**Stressor reactivity (SR) score.** The normative E-P relationship was calculated by fitting a mixed linear regression model of P against E across the entire sample and all time points. Adding a quadratic term did not improve model fit significantly ($\chi = 1.28$ (df = 1), $p = 0.259$). A random intercept and slope were added to the model. Individual SR scores were then calculated as the residuals at each time point to the E-P line. Variance explained ($R^2$) was calculated following the approach by Nakagawa[76], which estimates the variance explained by all effects (conditional $R^2$) and the variance explained only by fixed effects (marginal $R^2$).

### Analyses

**Intervention effects..** Analyses of intervention effects on E, P, SR, and PASS-content followed the protocol paper but were not pre-registered[36]. For all effects estimations, we used an intention-to-treat (ITT) approach considering T3 (follow-up) as the primary endpoint. Each effect was tested in a linear mixed model that included participant as random effect and covariates as fixed effects, as well as time, group, and time by group effects. Covariate selection is not detailed in the study protocol; for consistency with MARP and LORA, we used the same covariate selection procedure as there (see above and Supplementary Table 18). The protocol paper specifies that the analysis of the primary outcome (PHQ-ADS, that is, P, as already reported in the primary analysis[40]) and of all secondary outcomes (that is, here, SR) should use baseline (T0) values of the given outcome as covariate (one baseline-adjusted model and one fully adjusted model including also other covariates). For consistency with the analyses of E and PASS-content effects, the P and SR analyses reported in the main text and Fig. 6 do not include baseline measures among the covariates. Baseline-adjusted results are given in Supplementary Table 16a.

**Prediction analysis (association of baseline PAS with SR)..** To test the effect of the baseline (T0) PASS-content score on SR, we computed a linear mixed model with a random effect for time and a nested random intercept for group and participant. This reflects that participants

within a group might be more similar to each other and that they can only belong to one randomization group. Covariates (age, gender, education) were included as fixed effects. This analysis was not prespecified.

**Mediation analyses..** Post-hoc power analyses were conducted utilizing the shiny extension by Schoemann and colleagues[77]. A monte Carlo power analysis for an indirect effect >0 (1000 repetitions, 20,000 Monte Carlo draws per repetition) indicated power of 0.8 would be reached at a sample of $N = 170$, power of 0.9 at $N = 210$. In the analysis of the sample of complete cases ($n = 135$) the power reached is 0.69.

All analyses comprise secondary analyses on pre-specified exploratory outcomes, which were outlined in the protocol paper released before recruitment started[36] but not in the trial registry. The protocol paper specifies that we should investigate mediation of the intervention effect on SR by PASS-content, but does not prescribe a specific method. Our main analyses used the VanderWeele approach (mediation based on potential outcomes) and were performed with five imputed datasets using direct counterfactual imputation estimation. Results are presented in additive scale. We fit two regression models, the intervention-outcome model and the mediator-outcome model[79,80]. The VanderWeele approach allows for estimating the effect of the intervention mediated through PASS-content(T2) by comparing the potential outcomes: (i) the observed intervention condition and (ii) the potential (and unobservable) outcome of an intervention condition in which the intervention's effect on SR(T3) through PASS-content(T2) is blocked[81] (see Supplementary Fig. 2F).

Here, natural effects (nde) are the effects of the intervention on the outcome (SR) if the mediator (PASS-content) is allowed to arise naturally and is not set at a specific value. The nde is the effect the intervention would have if it did not produce changes in PASS-content. nde represents the difference between the two potential outcomes. The first potential outcome is the risk of higher SR if every participant received the intervention, and their PASS-content score would be as if they had not received it. The second (observed) potential outcome is the risk of higher SR if every participant received the intervention and their PASS-content score would be the value that it would take if they had received the intervention. Therefore, nde indicates how SR changes if the intervention's effect on SR through PASS-content is blocked[82,83]. The natural indirect effect (nie) compares two potential outcomes. The first and observed outcome would be the SR scores if every participant were exposed to the interventions. The second potential outcome would be SR scores if every participant received the DWM and PM + , but PASS-content would remain as if they had not received the interventions.

Age, gender, and education level as well as SR and PAS at baseline (T0) were included as covariates. The main results presented include causal effect estimation based on counterfactual imputation of ten datasets. Further sensitivity analyses were computed, including models allowing for exposure-mediator interaction and using weight-based estimation, rather than regression-based estimation. For ease of interpretation, these are not reported in the main text, as they all rendered similar confidence intervals. For the comparison of different models, see Supplementary Table 19. The SR scores were scaled prior to the effect estimation to take values from 0 to a 100, so that the difference scale can be interpreted.

We report the total effect of the intervention, the natural direct effect (nde), the natural indirect effect (nie), and the proportion mediated (pm). The nde is the effect the intervention would have if it did not produce changes in PAS[82,83]. The nie is the effect of the intervention through PAS. For both nde and nie, a pure and a total effect is estimated, reflecting which of the effects absorbs a potential interaction between the mediator and the group. We report the pure indirect effect and the total direct effect.

## Data availability

Pre-processed sum scores for each instrument used in this study, disaggregated by participant, have been deposited in the Open Science Framework repository https://doi.org/10.17605/OSF.IO/D4EXM. Item-level responses to all instruments, except those used to assess stressor exposure and sociodemographic information, disaggregated by participant, are also available from OSF. These data are available without restriction. Individual-level information on sociodemographic variables and item-level stressor exposures are not publicly available, as these could potentially allow for the re-identification of participants and thereby compromise data anonymization under the European Union's General Data Protection Regulation (GDPR). As a result, the publicly available minimum datasets support the replication of crude (unadjusted) models only. Adjusted analyses requiring demographic variables or item-level exposure data cannot be reproduced without additional data access. In accordance with the study protocols and informed consent provisions, access to pseudonymized raw data is restricted to members of the consortia. Readers may direct requests for raw data analyses by consortia members to **rkalisch@uni-mainz.de** (for MARP and LORA) and **joseluis.ayuso@uam.es** (for RESPOND-RCT Spain). We aim to respond to such requests within 4 weeks and will work with the requesting party to determine how and under what terms these analyses can be conducted. Priority will be given to analyses that aim to reproduce the current findings or are closely aligned with the original study objectives. Some of the data used in this manuscript have been analyzed in previous publications with the following PubMed identifiers (PMIDs): 38585485, 38306328, 35577829, 32419371, 32130154, 31557545 (MARP), 39531708, 39509222, 38306328, 37334645, 36130942, 34622343, 34539510, 34282129, 33432755, 32683526 (LORA), and 37263708 (RESPOND-RCT).

## Code availability

Statistical code, data dictionaries, and readme files are available on https://doi.org/10.17605/OSF.IO/D4EXM

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

## Acknowledgements

This project has received funding from the European Union's Horizon 2020 research and innovation program under Grant Agreement numbers 777084 (DynaMORE project, grant holders: O.T., K.S.L.Y., R.K.) and 101016127 (RESPOND project, grant holders: Co.B., K.L., D.M., M.M., M.S., O.T., J.M.H., J.L.A.-M., R.K.), Deutsche Forschungsgemeinschaft (grant number: Grant CRC 1193, subproject Z03, grant holders: K.L., A.R.), the Stifung of Rhineland-Pfalz für Innovation (grant number: 961-386261/1080, grant holders: K.L., O.T., R.K.), the German Research Foundation (DFG) (LORA-Risk, grant number: 519365575, grant holder: M.M.P., A.R.), the Technische Universität Dresden (Maria Reiche Postdoctoral Fellowship, grant holder: L.P.), and the Instituto de Salud Carlos III (grant number: CD22/00061, grant holder: R.M.).

## Author contributions

P.P.-R.: Conceptualization, Methodology, Software, Validation, Formal analysis, Data curation, Writing—Original draft, Visualization. R.M.: Conceptualization, Methodology, Validation, Investigation, Data Curation, Writing—Original Draft, Project Administration. A.R.-H.: Methodology, Validation, Writing—Original draft. L.M.C.P.: Methodology, Software, Investigation, Data curation, Writing—

Review and Editing. M.Z.: Investigation, Data curation, Writing—Review and Editing. K.F.A.: Investigation, Data curation, Writing—Review and Editing. Co.B.: Conceptualization, Writing—Review and Editing, Funding acquisition. U.B.: Conceptualization, Writing—Review and Editing, Supervision, Funding acquisition. Ca.B.: Conceptualization, Investigation, Writing—Review and Editing. A.C.: Methodology, Supervision, Writing—Review and Editing. M.F-N: Conceptualization, Investigation, Resources, Writing—Review and Editing, Project administration. B.K.: Investigation, Data curation, Writing—Review and Editing. K.L.: Conceptualization, Funding acquisition, Supervision, Project administration, Writing—Review and Editing. D.M.: Conceptualization, Writing—Review and Editing, Funding acquisition. K.R.M.: Conceptualization, Investigation, Resources, Writing—Review and Editing, Project administration. M.M.: Conceptualization, Writing—Review and Editing, Funding acquisition. A.M.-S.: Conceptualization, Investigation, Writing—Review and Editing. R.J.N.: Investigation, Data curation, Writing—Review and Editing. A.-L.P. Conceptualization, Writing—Review and Editing. M.M.P.: Project administration, Supervision, Writing—Review and Editing. M.P.: Conceptualization, Writing—Review and Editing. A.R.: Conceptualization, Funding acquisition, Supervision, Project administration, Writing—Review and Editing. C.S.: Investigation, Data curation, Writing—Review and Editing. An.S: Project administration, Investigation, Data curation, Writing—Review and Editing. A.l.S.: Conceptualization, Supervision, Writing—Review and Editing. M.S.: Conceptualization, Writing—Review and Editing, Funding acquisition. P.S.: Writing—Review and Editing. O.T.: Conceptualization, Funding acquisition, Supervision, Project administration, Writing—Review and Editing. A.W.: Conceptualization, Writing—Review and Editing. M.W.: Conceptualization, Funding acquisition, Supervision, Writing—Review and Editing. K.S.L.Y.: Methodology, Data curation, Project administration, Supervision, Writing—Review and Editing. J.M.H.: Conceptualization, Methodology, Resources, Writing—Review and Editing, Project administration, Funding acquisition. J.L.A.-M.: Conceptualization, Methodology, Resources, Writing—Review and Editing, Supervision, Project administration, Funding acquisition. R.K.: Conceptualization, Funding acquisition, Methodology, Project administration, Supervision, Validation, Visualization, Writing—Original draft.

## Competing interests

R.K. has received advisory honoraria from JoyVentures, Herzlia, Israel. K.F.A. has received remuneration from Janssen for consultancy services. The remaining authors declare no competing interests.

## Additional information

¹Leibniz Institute for Resilience Research, Mainz, Germany. ²Department of Psychiatry, Universidad Autónoma de Madrid (UAM), Madrid, Spain. ³Centro de Investigación Biomédica en Red en Salud Mental (CIBERSAM), Instituto de Salud Carlos III, Madrid, Spain. ⁴Instituto de Investigación Sanitaria del Hospital Universitario La Princesa (IIS-Princesa), Madrid, Spain. ⁵Department of Epidemiology, Columbia University Mailman School of Public Health, New York, NY, USA. ⁶Epidemiology Group, National School of Public Health, University of Antioquia, Medellín, Colombia. ⁷Clinical Psychology and Behavioural Neuroscience, Faculty of Psychology, Technische Universität Dresden, Dresden, Germany. ⁸Neuroimaging Center (NIC), Focus Program Translational Neuroscience (FTN), Johannes Gutenberg University Medical Center, Mainz, Germany. ⁹Department of Psychiatry, Psychosomatic Medicine and Psychotherapy, University Hospital Frankfurt, Frankfurt, Germany. ¹⁰Cooperative Brain Imaging Center - CoBIC, Goethe University Frankfurt, Frankfurt, Germany. ¹¹WHO Collaborating Centre for Research and Training in Mental Health and Service Evaluation, Department of Neurosciences, Biomedicine and Movement Sciences, Section of Psychiatry, University of Verona, Verona, Italy. ¹²Department of Psychology, RPTU University of Kaiserslautern-Landau, Landau, Germany. ¹³Department of Psychiatry, Clinical Psychology, and Mental Health, Hospital Universitario La Paz, Madrid, Spain. ¹⁴Instituto de Investigación Sanitaria del Hospital Universitario La Paz (IdiPAZ), Madrid, Spain. ¹⁵Faculty of Social Work, Health Care and Nursing Science, Esslingen University of Applied Sciences, Esslingen, Germany. ¹⁶Group of Epidemiology of Mental Disorders and Ageing, Sant Joan de Déu Research Institute, Esplugues de Llobregat, Barcelona, Spain. ¹⁷Research, Teaching, and Innovation Unit, Parc Sanitari Sant Joan de Déu, Sant Boi de Llobregat, Barcelona, Spain. ¹⁸Central Institute of Mental Health, Department of Neuropsychology and Psychological Resilience Research, Mannheim, Germany. ¹⁹Department of Psychiatry and Psychotherapy, Johannes Gutenberg University Medical Center, Mainz, Germany. ²⁰Care Policy and Evaluation Centre, Department of Health Policy, London School of Economics and Political Science, London, UK. ²¹Sorbonne Université, INSERM, Institut Pierre Louis d'Épidémiologie Et de Santé Publique (IPLESP), Équipe de Recherche en Épidémiologie Sociale (ERES), Paris, France. ²²Department of Public Mental Health, Central Institute of Mental Health, Medical Faculty Mannheim, Heidelberg University, Mannheim, Germany. ²³Department of Clinical, Neuro- and Developmental Psychology, WHO Collaborating Center for Research and Dissemination of Psychological Interventions, Amsterdam Public Health Research Institute, Vrije Universiteit Amsterdam, Amsterdam, The Netherlands. ²⁴Department of Epidemiology and public health, Sciensano, Brussels, Belgium. ²⁵Institute of Health and Society (IRSS), Université catholique de

Louvain, Brussels, Belgium. [26]Department of Psychiatry, Psychotherapy and Psychosomatic Medicine University Medicine Halle (Saale), Martin Luther University Halle-Wittenberg (MLU), Halle (Saale), Germany. [27]Partner site Halle-Jena-Magdeburg, German Center for Mental Health (DZPG), Halle (Saale), Germany. [28]DKFZ Hector Cancer Institute at the University Medical Center Mannheim, Mannheim, Germany. [29]German Cancer Research Center (DKFZ) Heidelberg, Division of Cancer Survivorship and Psychological Resilience, Heidelberg, Germany. [30]Department of Medicine, Universitat de Barcelona, Barcelona, Spain. [31]Department of Psychiatry, Hospital Universitario La Princesa, Madrid, Spain. [32]These authors contributed equally: José Luis Ayuso-Mateos, Raffael Kalisch. ✉e-mail: roberto.mediavilla@uam.es

