## [Transparent Peer Review file · Nature Communications]

Positive appraisal style predicts long-term stress resilience and mediates the effect of a pro-resilience intervention

Corresponding Author: Dr Roberto Mediavilla

Version 0:

Reviewer comments:

Reviewer #1

(Remarks to the Author)

The present manuscript reports results of three studies investigating the role of positive appraisal style (PAS) in the context of stress resilience. Findings of two independent longitudinal studies showed that PAS predicted resilience, i.e., over three and more years, and one of the studies also found PAS to mediate the relationship between positive effects and social support. Results of an interventional study showed that a multi-component intervention targeting various components of resilience factors increased PAS scores, and mediation analyses revealed that PAS scores were partly responsible for the effect on stress reactivity.

I think the authors are working on an important research line and I would like to compliment them on this group effort to run these theoretically- and clinically-relevant studies. Strengths of the investigation are the transdiagnostic feature of the examined processes and using both longitudinal and interventional designs. However, I also have a number of concerns, see for a summary below:

1. I very much agree with the authors that mediation analyses can offer important causal-mechanistic insights into the relationship between certain variables. However, such analyses are also limited in terms of what we can conclude, and in the present context I am not sure whether the used designs and analyses support the causal claims the authors present. That is, none of the studies directly manipulated appraisals in a controlled experimental setting. Further, not all analyses were significant and partly exploratory (e.g., MARP). Hence, when taking both these issues together, also in combination with the other analyses (i.e., the regressions), it seems not appropriate to have such a strong causal focus when it comes to the role of PASS (and see also point 2 and 3 to support this point from other angles).
2. The interventions applied in the RCT were quite heterogenous and I am wondering in how far all included processes are truly relevant in the context of appraisals. To illustrate, Problem Management Plus (PM+) included problem-solving techniques and strategies for promoting social support, and Doing What Matters (DMW) included strategies such as acting on values and being kind. I am aware of the argument the authors provide in the introduction (page 5, bottom page), i.e., that "PAS may be enhanced even by interventions that do not target it directly" but this lack of specificity also limits the results' interpretation. I totally see the relevance of these strategies in relation to the primary aim and analyses of the RCT, i.e., reducing mental health problems, but I miss a concrete and specific link in relation to the present research questions.
3. I would like to compliment the authors on being very transparent about the RCT analyses representing secondary analyses. I had a look at the RCT protocol paper and could find a description of the PASS and using it in the general context of mediation analyses (section data analysis). However, I could not find any specific analyses plans in the trial registration on ClinicalTrials.gov – maybe I overlooked it but did the authors pre-register the secondary analyses conducted in the present manuscript using the PASS data?

A similar question appears for the MARP and LORA data. I understand that this data was extracted from larger scale projects and assessment waves. However, there are bits of information in the present manuscript that feel slight puzzling and made me wonder what was planned a priori and what not. To illustrate, for the MARP data, a data freeze was performed in November 2022, what informed this data freeze at that specific time-point? Then it was stated that the mediation analyses in MARP was underpowered and exploratory, was this thus a post-hoc analysis? In contrast, the mediation analyses for LORA were not underpowered and not exploratory? I also had a look at reference 20 and 70 (MARP, Methods, page 20) to be able

to better understand the present data and analyses in the context of the larger project, aims, and analyses. However, Kampa et al., (preprint) seems to summarize the feasibility of a behavioral and functional magnetic resonance imaging (fMRI) battery in N = 55 healthy participants, and Kampa et al. (2024) describes a fMRI replication study in the context of stress resilience mechanisms using two independent samples (N = 54 and N = 103). For LORA, I had a look at reference 76 (Methods, page 23), but Chmitorz et al., (2020) merely reports the validation of the MIMIS questionnaire. Hence, these references did not provide an overarching study protocol for either project, which I expected, and thus did not help to extract the specific research question, hypotheses, and analyses presented in the present manuscript from the overall umbrella projects and methods used (but I may have checked the wrong sources?).

To summarize, I am partly struggling to understand how to interpret the present results best, i.e., what was planned in advance, what is a kind of primary outcome and what should be regarded as secondary/additional outcomes, what was exploratory/post-hoc etc.? Having more specific information about these issues, in combination with the overall aims of the corresponding umbrella projects, would clearly help to make a distinction between main versus supporting results, and, in turn, would help to get a general idea of what we can specifically learn from the present extracted data in the context of the general projects.

4. I may have missed that information but given the purpose of the present study I would recommend to also present internal consistency data for the measures used, especially the PASS scales.

5. I think the discussion would benefit from adding some more in-depth elaborations and explanations. To illustrate, LORA and MARP describe two very different samples and I think it would be good to include some thoughts about this and the rationale to combine and compare these two samples. It would be also helpful to present some more (potentially hypothetical) elaborations about the underlying processes driving the observed effects, especially given the manuscript's mechanistic focus. Paragraph 2 partly does this, but I would recommend to add some more concrete thoughts using the specific findings that are reported. To illustrate, what are the (different) mechanisms underlying PASS being a mediator for social support and SR versus stress recovery being a mediator for PASS and SR? What do the reported findings add to the existing data (see e.g., introduction page 5, middle page, stating "that negative associations of the well-established resilience factor perceived social support with SR were mediated by PASS-process)? How do the present findings inform experimental follow-up work to test specific, potential causal relationships? Finally, the discussion partly repeats information of the introduction, e.g., the first paragraph.

6. I sometimes found the manuscript very difficult to read and some aspects were presented in a rather abstract manner. Accordingly, I would like to invite the authors to re-check whether certain aspects could be presented more clearly. To illustrate, the second paragraph of the introduction compares different appraisal styles and their effects on mental health. For an unknown reader, the difference between a "realistic to mildly illusionary positive appraisal" versus an "overly (delusionally) positive appraisal tendency" may not be straightforward to understand and e.g., concrete examples would be helpful. Another example is at the end of the first page of the introduction: "By focusing on appraisal contents, rather than on the cognitive processes that can generate these contents (as in PASS-process), the PASS-content scale more directly targets the PAS construct." This sounds very abstract and also here an example or a more concrete writing style would help. A final example I would like to highlight comes from the discussion, where the authors speculate about the potential underlying processes PASS-process versus PASS-contents instruments target (i.e., unconscious vs. conscious). I think I understood the general idea, but it still may be helpful to critically re-check this section as it comes a bit out of the blue.

This points also relates to some more general issues, e.g., I found out very late that the included moderators of the first two studies were selected based on previous analyses of the corresponding data sets. In addition, I think the analyses could be presented clearer, including the results that are presented in Table 1 and 2, e.g., I do not understand what the two columns within each column represent and it seems that the order of the columns are not presented in a chronological manner (which is fine but this might have a reason?). In addition, I am not sure whether I understand the general analytical set-up - the analyses seem to merely include predictors assessed at baseline and therefore represent a "prediction from baseline" analysis and not a crossed-legged analysis which would also control for certain variables from other time-point than baseline (and see point 3)?

7. It would be interesting to also link the present data with some mental health data, which should be available in case I understood the methods section correctly, e.g., testing whether different operationalizations of appraisals (i.e., process vs. content) are related with depression and anxiety scores / predict these scores.

8. I may have overlooked this information but does the OSF project include a data codebook explaining all the variables included in the corresponding data files? The README file included some general information and I could not find anything in the other files?

(Remarks on code availability)

Due to my upcoming holidays I was not able to review the code in much detail / repeat and check the analyses. I had a look at the files and in case I understood everything correctly there is no codebook describing the various variables in the data files.

Reviewer #2

(Remarks to the Author)

The paper includes several studies testing the potential causal effects of PAS on stress reactivity. It concerns quite a few impressive sets of data and the evidence for the potential causal role of PAS in resilience is convincing, which might have interesting potential for future studies on the prevention of exaggerated stress related responses.

The paper is quite an information rich paper, that is it includes a lot of information as it describes several (large) studies. Because data of three studies is used, a lot of information is also not provided, in order to make it fit in one paper, which makes it slightly rough to read and hard to wrap your head around. I think the paper makes a contribution to the field of psychology and might lead to more studies on this topic to further out insight in the concept of PAS and the role it could play in the stress resilience.

I have a few questions that I would like the authors to respond to or / and address:

Minor question: why was in the MARP study a inclusion criterion used of three prior negative life events, specifically 3 (why not 1, or 2?), a rationale would be helpful to understand why this number was chosen.

When introducing the RESPOND-RCT Spain is introduced in the introduction, it is explained that the first intervention step includes an ACT approach, the second step seems to rely on CBT interventions. It makes me curious and it would be interesting to already mention perhaps whether ACT and CBT had a differential effect on PAS. At least this question raises in my head while reading about the study, and it would be nice to mention some findings in the introduction already (or perhaps in the discussion when going into options to modify PAS).

Results: Pass-content shows full mediation in the LORA study, only partial mediation in the MARP study. Doesn't that mean that perhaps both social support and PASS-content seem to be important to focus on with the eye on interventions?

PAS seems to be almost the opposite of repetitive negative thinking (RNT), also a well known risk factor of psychopathology and stress responses. I was wondering what the view of the researchers is on this and how they think it relates to RNT (how does it differentiate from it perhaps?).

Although the results include a subanalysis when testing the longitudinal association between PASS and stress reactivity with only the most stress-exposed individuals (p.8). This is encouraging as it seems to support the view that the level of exposure does not influence the found longitudinal relationship. Also in the discussion the authors argue that "PAS appears to protect against various types of stressors" and "participants experienced qualitatively different exposure" (p.18). Although I agree with this statement, based on the data that has been provided, I do wonder whether it also generalizes to more extreme stressors including traumatic events. Looking at the descriptives, most stressors that have been reported include common and relatively "mild" stressors. Even though I believe also here the dimensional model applies, it remains to be tested. Especially since traumatic events can shake one's beliefs about the self, others and the world drastically. I would like the authors to elaborate a bit more on this discussion on generalizability to more severe or clinical contexts.

(Remarks on code availability)

I did not run the code but data seems available

Reviewer #3

(Remarks to the Author)

The authors present a compelling argument for the importance of Positive Appraisal Style (PAS) as a crucial factor in stress resilience. The authors combine data from three studies – two longitudinal observational studies (MARP, LORA) and a randomized controlled trial (RESPOND-RCT Spain) – to demonstrate the predictive validity, mediating role, and modifiability of PAS. The results are generally strong and well-aligned with existing literature on resilience, offering valuable insights into potential pathways for bolstering mental health in the face of adversity.

The key attributes of this paper are the following:

Prospective evidence for PAS as a resilience factor: The prospective association between PAS and lower stressor reactivity (SR) across extended periods in both MARP and LORA convincingly establishes PAS as a robust predictor of resilience.

This builds upon and significantly extends prior work that primarily focused on cross-sectional or short-term prospective associations.

Demonstration of mediation: The authors demonstrate that PAS mediates the relationship between perceived social support and SR in both observational studies. Additionally, the finding that perceived good stress recovery mediates the PAS-SR link aligns with the proposed mechanism of optimized stress response regulation through positive appraisals. This provides further support for the theoretical framework presented by PASTOR.

Intervention-induced changes in PAS: The results from the RESPOND-RCT Spain are particularly noteworthy, showing that a multi-component intervention targeting a broad range of resilience factors leads to increased PAS, which in turn mediates improvements in SR. This strongly suggests a causal role for PAS in resilience, highlighting its potential as a target for therapeutic interventions.

Comprehensive and methodologically rigorous approach: The authors utilize a comprehensive approach, combining data from multiple studies with diverse populations and stressors. The use of validated instruments, residualized SR scores to control for exposure differences, and sophisticated mediation analyses strengthens the robustness of their findings.

Here are some more detailed issues to address upon r&r:

The manuscript could benefit from a more explicit discussion of the relationship between PAS and other theoretical models of resilience, particularly those emphasizing flexible coping and emotion regulation (e.g., Bonanno's work, emotion regulation flexibility). How does this relate to existing models? Grounding this work in the larger research area may be warranted

The discussion of methodological limitations related to self-report measures of PAS is insightful. However, the authors could expand on potential strategies to address these limitations in future research, such as incorporating objective measures of appraisal (e.g., physiological, implicit cognitive tasks) alongside self-report. What types of measures are both scalable and more precise and mechanistic than self-report?

Specificity of the intervention: While the multi-component nature of the RESPOND-RCT intervention is acknowledged, the authors could further elaborate on the specific components and their potential mechanisms of action. This would help clarify how the intervention might have targeted different pathways leading to PAS enhancement.

Alternative explanations for intervention effects: The manuscript could benefit from a more detailed discussion of potential alternative explanations for the observed intervention effects. For example, could changes in stressor exposure (as indicated by the group by time interaction effect on E) have directly contributed to SR improvements, independent of PAS? Addressing such possibilities would strengthen the causal inferences drawn from the RCT data.

Theoretical implications: The manuscript could benefit from a more comprehensive discussion of the broader theoretical implications of the findings for understanding resilience processes. How does PAS interact with other known resilience factors? Does PAS primarily operate at the level of primary appraisal (initial assessment of threat) or secondary appraisal (evaluation of coping resources)? Addressing these questions would enhance the theoretical contribution of the study.

(Remarks on code availability)
N/A

Version 1:

Reviewer comments:

Reviewer #1

(Remarks to the Author)

I would like to thank the authors for their hard work on the manuscript and for their thorough responses to the comments. As highlighted by Reviewer 2, this paper is rich in information and analyses. The revisions have significantly enhanced the quality of the work, and I believe this manuscript will make a valuable contribution to the literature.

(Remarks on code availability)

Reviewer #2

(Remarks to the Author)

Thank you for your detailed responses to my comments and questions. Although I think that the authors have adequately addressed my questions, I do not find the manuscript with track changes, and it is therefore difficult to assess the extent to which all comments (including those of the other reviewers) have been addressed in the manuscript itself. Especially since this is still a very information-rich article that is not easy to read, I would like the authors to upload the manuscript with track changes for reasons of efficiency. If the authors have done this (I can't seem to find it), I would be happy to receive it from the editorial office, and then I can judge whether I'm satisfied with the revision.
Thank you very much!

(Remarks on code availability)
will fill this out later

Reviewer #3

(Remarks to the Author)

(Remarks on code availability)

I am satisfied that the authors have addressed my key concerns. Any lingering concerns are more stylistic than substantive.

REVIEWER COMMENTS

Reviewer #1 (Remarks to the Author):

The present manuscript reports results of three studies investigating the role of positive appraisal style (PAS) in the context of stress resilience. Findings of two independent longitudinal studies showed that PAS predicted resilience, i.e., over three and more years, and one of the studies also found PAS to mediate the relationship between positive effects and social support. Results of an interventional study showed that a multi-component intervention targeting various components of resilience factors increased PAS scores, and mediation analyses revealed that PAS scores were partly responsible for the effect on stress reactivity.

I think the authors are working on an important research line and I would like to compliment them on this group effort to run these theoretically- and clinically-relevant studies. Strengths of the investigation are the transdiagnostic feature of the examined processes and using both longitudinal and interventional designs.

→ REPLY: We are especially grateful to reviewer 1 for their thorough analysis of our paper and the detailed feedback, which we hope they will agree has allowed us to considerably improve the manuscript. We would like to add a small comment that PAS was found to mediate the relationship between social support and resilience not only in one of the observational longitudinal studies but in both of them (MARF and LORA), that is, in all studies in which social support data were available.

However, I also have a number of concerns, see for a summary below:

1. I very much agree with the authors that mediation analyses can offer important causal- mechanistic insights into the relationship between certain variables. However, such analyses are also limited in terms of what we can conclude, and in the present context I am not sure whether the used designs and analyses support the causal claims the authors present. That is, none of the studies directly manipulated appraisals in a controlled experimental setting. Further, not all analyses were significant and partly exploratory (e.g., MARF). Hence, when taking both these issues together, also in combination with the other analyses (i.e., the regressions), it seems not appropriate to have such a strong causal focus when it comes to the role of PASS (and see also point 2 and 3 to support this point from other angles).

→ REPLY: We thank the reviewer for pointing us towards the limitations that exist in what we can conclude about the causality of positive appraisal style (PAS) for resilience on the basis of a mediation analysis with PAS, especially when there was no experimental manipulation of the mediator PAS (MARF, LORA) or the manipulation did not directly target the mediator PAS (RCT). To briefly explain, we chose to talk about causal effects in the original manuscript because we had carefully assessed confounding factors (MARF, LORA, RCT) and had indirectly manipulated our exposure (RCT) to

obtain causal estimates that were as unbiased as possible to answer our main research question, that is, whether PAS drives changes in resilience. In addition, in all three data sets, we had observed that PAS as measured with the PASS-content instrument (which is the more reliable and theoretically more valid measure for PAS than the PASS-process questionnaire, see discussion) significantly mediates the effects of social support on resilience (MARP, LORA) or the effect of the intervention on resilience (RCT). There were no non-significant mediation effects, and the exploratory effects in MARP for a mediation of social support effects on resilience were confirmed in the replication sample LORA.

However, we acknowledge that our causal claims are still limited by key constraints, such as unmeasured confounding and reverse causation biases (MARP, LORA) or indirect manipulation of the exposure (RCT). We therefore now refrain from causal claims throughout the manuscript and only discuss that our findings are in agreement with a potential causality of PAS for resilience, which however would have to be tested in future studies. Please see the adjustments in title (deletion of “causal-mechanistic”), abstract (“potential causal effects” instead of “likely causal effects”), and main text (P6, L18; P6, L41; P17, L24; P19, L43; P19,L43) in the revised version. Notably, we write on P19, L46: “We, however, emphasize that the intervention targeted a wide range of potential factors, which included appraisal but were not limited to this construct. We report largely unbiased estimates of causal effects of PAS on resilience. However, these estimates are still hindered by unmeasured confounding, reverse causation, or indirect manipulation of the exposure. Future intervention studies that directly manipulate PAS will help make stronger causal claims that strengthen theory and inform intervention development. These studies should focus on the PASS-content instrument as an intervention target and effect mediator, which in the present work is more reliable and more strongly associated with SR than PASS-process and also has higher construct validity (see above).”

2. The interventions applied in the RCT were quite heterogenous and I am wondering in how far all included processes are truly relevant in the context of appraisals. To illustrate, Problem Management Plus (PM+) included problem-solving techniques and strategies for promoting social support, and Doing What Matters (DMW) included strategies such as acting on values and being kind. I am aware of the argument the authors provide in the introduction (page 5, bottom page), i.e., that “PAS may be enhanced even by interventions that do not target it directly” but this lack of specificity also limits the results’ interpretation. I totally see the relevance of these strategies in relation to the primary aim and analyses of the RCT, i.e., reducing mental health problems, but I miss a concrete and specific link in relation to the present research questions.

→ REPLY: We apologize for a slightly lengthier response to this question, since it is so key to our investigations. A central and integral claim of positive appraisal theory of resilience (PASTOR) is that PAS mediates the effects of other resilience factors on resilience (Kalisch et al., 2015). This

claim is based on the idea that any resilience factor will ultimately benefit resilience because it shapes the way one typically appraises stressors (potential threats to one's goals or needs) in a more positive way, and this is eventually what helps people stay mentally healthy despite adversity. In Kalisch et al. (2015), section 4.2.4.2, we have given the explicit example of social support as a more "distal" resilience factor and have posited that a person who perceives themselves as being well supported will usually perceive stressors as less threatening, for instance because they know that their support network would provide them with coping resources if needed. We also there argue that social support may have the additional effect of reducing one's actual stressor exposure, because helpers may take some burden off one's shoulders (see Figure 4A in Kalisch et al., 2015). Critically, however, this latter effect is covered by our extensive stressor exposure monitoring and the inclusion of this variable into our outcome score (SR score), which corrects for individual differences in exposure. Provided such correction, the only remaining statistically visible effect of social support on resilience should be via appraisal, according to the theory (Figure 4B in Kalisch et al., 2015). The example illustrates why and how PAS is posited to be a "proximal" resilience factor, that is, an immediate causal factor in the effect path towards resilience and why and how it is thought to integrate the effects of other resilience factors.

Analogous arguments apply to other resilience factors than social support (Kalisch et al., 2015, 4.2.4.2 uses the examples of life history and genotype). To address the other potential resilience factors proposed by the reviewer as being positively affected by the PM+/DWM intervention in our third study in the paper: If one has better problem-solving techniques at one's disposal, this will a) shape one's appraisal of one's coping potential positively (enhancing PAS), and it will b) reduce stressor exposure because problems get solved more efficiently (the effect on exposure being factored out via the SR score). If one learns to act more in agreement with one's values and is kinder, this will a) reduce the perception of one's own actions as threats to one's self-esteem (one of the strongest stressors for most people) and hence enhance PAS, and it may b) also increase or decrease actual stressor exposure (e.g., increase because one avoids social conflicts less), but this will in any case be factored out through the SR score.

In sum, our theory predicts an effect of any intervention on resilience via PAS, no matter what the exact ingredients of the intervention are.

Therefore, the present intervention trial is relevant to the question of how stressor appraisal and resilience are related.

Please note that effect mediation by PAS of social support effects on resilience have so far been observed in all studies in which both social support and PAS data were available (MARP and LORA in this paper, further: Bögemann et al., 2023; van der Heide et al., 2024a; Veer et al., 2021; Zerban et al., 2023). The effect is thus apparently very robust.

We realize that we were not explicit enough in the paper and that there is a need to explain the relevance of the intervention trial more. For space reasons, we are doing this in the form of an extended discussion in a supplement, to which we refer on P6, L17 of the introduction. Please also note that reviewer 3 has asked a similar question, which we also address with these additions to the manuscript.

3. I would like to compliment the authors on being very transparent about the RCT analyses representing secondary analyses. I had a look at the RCT protocol paper and could find a description of the PASS and using it in the general context of mediation analyses (section data analysis). However, I could not find any specific analyses plans in the trial registration on ClinicalTrials.gov – maybe I overlooked it but did the authors pre-register the secondary analyses conducted in the present manuscript using the PASS data?

→ REPLY: The reviewer is correct. The specific PAS, SR score, and mediation analyses were not preregistered in the trial registration on ClinicalTrials.gov. That registration focused only on the primary analyses. As we write on P14, L13 of the original manuscript: “All secondary analyses reported in this paper were performed ..., as specified in the study protocol (Mediavilla et al., 2022).” The study protocol paper (Mediavilla et al., Dig Health 2022) was submitted January 17, 2022, while the study was ongoing, and accepted September 11, 2022, shortly after the termination of data collection. Hence, there was no formal preregistration of the secondary analyses reported in the current paper, only pre-specification, as described.

A similar question appears for the MARP and LORA data. I understand that this data was extracted from larger scale projects and assessment waves. However, there are bits of information in the present manuscript that feel slight puzzling and made me wonder what was planned a priori and what not. To illustrate, for the MARP data, a data freeze was performed in November 2022, what informed this data freeze at that specific time-point?

→ REPLY: There was no specific scientific rationale for performing the data freeze at this time point, except that at this time point we felt that there were enough data accumulated to permit analyses of PAS effects (time point B2 was finished in all participants, while completion of B3 would take another 12 months, plus we were uncertain whether there would be a sufficient number of participants left after B3 for meaningful analyses anyway). The most important motivation was, however, purely logistic and related to staff availability. At this stage, the processing of the B3 data is still ongoing, and the number of participants providing B0 and B3 data is projected to be around 70, suggesting that an inclusion of B3 data would not benefit the current analysis. Please note that all MARP analyses are clearly labelled as exploratory (“discovery sample”). We realize that the wording “...these finding confirm a link between positive appraisal style and resilience ...” on P8, L39 of the original manuscript was misleading and have now replace “confirm” with “suggest”.

Then it was stated that the mediation analyses in MARP was underpowered and exploratory, was this thus a post-hoc analysis? In contrast, the mediation analyses for LORA were not underpowered and not exploratory?

→ REPLY: We apologize for not pointing out our analysis strategy sufficiently clearly in the original manuscript. At the start of MARP, we had not yet

validated the two PAS questionnaires (Petri-Romão et al., 2024) and we had not yet developed the methods to calculate the outcome score (SR score) in detail (Kalisch et al., 2021). Therefore, yes, although we planned from the start of MARP to use this data set to test PASTOR (as stated in the preprint Kampa et al., 2018), the exact analyses in MARP are post hoc and exploratory, which is why we designated the MARP sample as “discovery sample”. Their specific set-up has also been shaped by the four studies published in 2023 and 2024 that we cite in the manuscript (Bögemann et al., 2023; van der Heide et al., 2024a; Veer et al., 2021; Zerban et al., 2023) and that have used a PAS questionnaire and SR as outcome score. We now make the exploratory nature of the MARP analysis more explicit on P8, L42: “Note that, although the MARP study was designed among others to provide data to test , the exact testing methods were not prespecified at the start of the study and the PASS instruments and the analytical strategies were developed during study conduct based on experiences with MARP and other studies (see introduction)(Bögemann et al., 2023; Kalisch et al., 2015; Petri-Romão et al., 2024; van der Heide et al., 2024a; Veer et al., 2021; Zerban et al., 2023). Hence, all analyses of MARP data reported here are of exploratory nature and require independent replication.”

We felt obliged to inform the reader that the mostly significant mediation results stemmed from underpowered analyses, and hope that this adds to the transparency of our report.

Concerning LORA, yes, based on the results of MARP, LORA was used as a replication sample, and the LORA analyses were not exploratory and also turned out to be well powered, as stated. Appended to above statement on the nature of MARP analyses, we now state on P9, L1: “The following analyses of the LORA sample serve this purpose and were set up to match the analyses of the MARP sample as closely as possible.”

I also had a look at reference 20 and 70 (MARP, Methods, page 20) to be able to better understand the present data and analyses in the context of the larger project, aims, and analyses. However, Kampa et al., (preprint) seems to summarize the feasibility of a behavioral and functional magnetic resonance imaging (fMRI) battery in N = 55 healthy participants, and Kampa et al. (2024) describes a fMRI replication study in the context of stress resilience mechanisms using two independent samples (N = 54 and N = 103). For LORA, I had a look at reference 76 (Methods, page 23), but Chmitorz et al., (2020) merely reports the validation of the MIMIS questionnaire. Hence, these references did not provide an overarching study protocol for either project, which I expected, and thus did not help to extract the specific research question, hypotheses, and analyses presented in the present manuscript from the overall umbrella projects and methods used (but I may have checked the wrong sources?).

→ REPLY: In further extension of our above reply, there is not yet a study protocol for MARP, and we therefore cited Kampa et al. (2018 preprint), and Kampa et al. (2020) to give the reader the available information about the study. The most detailed and authoritative information to date, however, is the methods section of the current paper. In the LORA study, data collection

started around the same time as MARP and, hence, as for MARP, there was no specific analysis protocol for testing PASTOR ready at that time. The now available LORA study protocol (Chmitorz et al., 2020) therefore only mentions a) the general idea of mediation of many resilience factors by more “proximal” ones (citing the Kalisch et al., 2015, paper on PASTOR), b) Positive Appraisal Style as baseline predictor, and c) the intention to use the mental health and stressor exposure data in this sample to develop methods to accurately quantify resilience (which resulted in the SR score method used here).

Note that we do not claim anywhere in the manuscript that the MARP and LORA analyses were preregistered or prespecified. The specific research questions, hypotheses, and analyses in the present manuscript have been developed based on methodological (Kalisch et al., 2021; Petri-Romão et al., 2024) and empirical (Bögemann et al., 2023; van der Heide et al., 2024a; Veer et al., 2021; Zerban et al., 2023) work done since the start of these studies and are explained in their development in the introduction.

To summarize, I am partly struggling to understand how to interpret the present results best, i.e., what was planned in advance, what is a kind of primary outcome and what should be regarded as secondary/additional outcomes, what was exploratory/post-hoc etc.? Having more specific information about these issues, in combination with the overall aims of the corresponding umbrella projects, would clearly help to make a distinction between main versus supporting results, and, in turn, would help to get a general idea of what we can specifically learn from the present extracted data in the context of the general projects.

➔ REPLY: We hope that our above clarifications and additions to the text have answered the reviewer’s questions.

Please further note that the general projects, as is inherent to the nature of large-scale many-collaborator projects (which would otherwise never happen), have not been designed to test only one specific theory or hypothesis.

4. I may have missed that information but given the purpose of the present study I would recommend to also present internal consistency data for the measures used, especially the PASS scales.

➔ REPLY: Internal consistency of PASS-process ($\omega=.85$; $\alpha=.78$) and PASS-content ($\omega=.88$; $\alpha=.87$) in a different sample of $N=3131$ had already been reported in the validation paper (Petri-Romão et al., 2024). We now also present internal consistency data for MARP (P22, L12 and P22, L21), LORA (P24, L34), and RESPOND-RCT (P25, L25) in the manuscript. They are similar.

5. I think the discussion would benefit from adding some more in-depth elaborations and explanations. To illustrate, LORA and MARP describe two very different samples and I think it would be good to include some thoughts about this and the rationale to combine and compare these two samples.

→ REPLY: We happily provide more in-depth elaboration. The difference between the MARP and LORA samples (and also to the RESPOND-RCT) in terms of their demographic composition as well as their stressor exposure are very useful, because they allow us to test at least to some extent whether PAS is a general, or universal, resilience factor, which would greatly increase its relevance. For the same reason, we also emphasize in the manuscript that the earlier studies (Bögemann et al., 2023; van der Heide et al., 2024a; Veer et al., 2021; Zerban et al., 2023) that had indicated preliminary PAS effects were done in yet other types of samples. See P18,L47, starting: “PAS is a resilience factor across different populations: ...”. We there also emphasize that the different samples have experienced different kinds of stressor. See P19,L6, starting: “A final insight is that PAS appears to protect against various types of stressors”. Altogether, we believe that the heterogeneity of our samples is a special strength of our paper and now explicitly say on P19, L14: “In sum, the available results suggest that PAS may be a universal resilience factor valid in different types of population and in different stressor situations.”

It would be also helpful to present some more (potentially hypothetical) elaborations about the underlying processes driving the observed effects, especially given the manuscript’s mechanistic focus. Paragraph 2 partly does this, but I would recommend to add some more concrete thoughts using the specific findings that are reported. To illustrate, what are the (different) mechanisms underlying PASS being a mediator for social support and SR versus stress recovery being a mediator for PASS and SR?

→ REPLY: We have tried to answer to this request by adding more specific thoughts into the introduction of the paper, where the theoretical basis of our hypotheses is explained and the expected mediations are already formulated (though not in sufficient detail in the original version of the manuscript). By elaborating and extending on the mediation hypotheses that are already mentioned there, we save space compared with extending the discussion. This also allows us to keep the repetition of information on the theory in the first paragraph of the discussion (see the reviewer’s next remark), which we felt is important to orient the reader back to the big questions of the paper after the extensive results report, to its present minimum. We hope the reviewer finds this is a good solution.

Concretely, concerning the idea of good stress recovery mediating the effects of PAS on resilience: On P4, L23, after having explained that individuals with a PAS are more likely to avoid unnecessary stress reactions or over-reactions, we add: “That is, their stress responses will on average be well regulated in terms of magnitude (no over-shoot in response amplitude), duration (fast recovery after stressor termination), and quality (appropriate response selection)^{2,14}.” And further below when summarizing the existing findings, we elaborate (P6, L3): “PASTOR also posits that positive appraisers show an optimal stress response profile, which eventually protects their mental health in adverse life situations^{12,14}. In line with that claim, we also observed that negative associations of PASS-process with SR were

mediated by a self-assessment of participants' stress recovery as one way to assess optimized stress response regulation, such that participants with a higher PAS also reported to more easily recover from stressors and showed correspondingly lower SR." In this way, the hypothetical mechanism leading from PAS to better resilience is already spelled out in the introduction. Please also note the concluding paragraph in the discussion (P20, L8): "... our findings allow us to start painting a coherent mechanistic picture of resilience, whereby factors that promote a tendency to appraise stressors in a mildly unrealistically positive fashion lead to optimally regulated (fine-tuned, situation-appropriate) stress responses, and this in turn preserves bodily and cognitive capacities that are crucial for long-term mental functioning."

Concerning the idea of PAS mediating the effects of social support on resilience, we hope that the explanations starting P5, L42 of the introduction ("PASTOR also claims that PAS is an integrative resilience factor that mediates...") in combination with the extended discussion in the supplement that we have there added (P18, L20) (in response to the reviewer's earlier question on how different resilience factors might be mediated by PAS) will fulfil the purpose of mechanistic elaboration.

What do the reported findings add to the existing data (see e.g., introduction page 5, middle page, stating "that negative associations of the well-established resilience factor perceived social support with SR were mediated by PASS-process)? How do the present findings inform experimental follow-up work to test specific, potential causal relationships? Finally, the discussion partly repeats information of the introduction, e.g., the first paragraph.

➔ REPLY: On the question of what our findings add to the existing data, we would like to point the reviewer to the third paragraph in the discussion, which uses the case of the observed associations of PAS with SR in the paper to explain the important new insights that these generate (P18, L42). These arguments analogously apply to the findings of mediation of social support effects on SR: "Our results show that PAS - measured with either instrument, but more so with PASS-content – prospectively predicts low stressor reactivity over long time frames of more than three years. This is an important insight, since the initial studies with PASS-process had either tested association with SR scores derived from a single, concurrent time point (asking participants about their stressor exposure and changes in mental health in the past weeks (Veer et al., 2021; Zerban et al., 2023)), or from short prospective intervals of five weeks (Bögemann et al., 2023) or six months (van der Heide et al., 2024b). Another insight from the available data is that PAS is a resilience factor across different populations: European adults (Bögemann et al., 2023; Veer et al., 2021), mental health practitioners from various countries (Zerban et al., 2023), Dutch Parkinson patients (van der Heide et al., 2024a), German young adults (MARF), German adults (LORA), and Spanish health care workers (RESPOND-RCT Spain). ... A final insight is that PAS appears to protect against various types of stressors. ... In sum, the available results suggest that PAS may be a universal resilience factor valid in different types of population and in different stressor situations."

On the question of how our findings inform follow-up, we have extended the last-but-one paragraph of the discussion, which already mentioned in the original manuscript the need to complement the questionnaire-based measurement of PAS with a more objective, task-based measure as well as the need to compare PASTOR with regulatory flexibility theories (P20, L17): “Another challenge will be to establish the generalizability of our observations to a broader range of populations and types of adversity to thus definitively establish PAS as a universal resilience factor.” Please note that there is now also a statement on the need to directly manipulate PAS experimentally in follow-up work, to thus establish causality on P20, L3 of the discussion, in reply to the reviewer’s earlier question.

We hope that the reviewer can agree with our decision to leave the first paragraph of the discussion as is, despite some repetition, as we believe that it is helpful for getting the reader back on track after the long results section.

6. I sometimes found the manuscript very difficult to read and some aspects were presented in a rather abstract manner. Accordingly, I would like to invite the authors to re-check whether certain aspects could be presented more clearly. To illustrate, the second paragraph of the introduction compares different appraisal styles and their effects on mental health. For an unknown reader, the difference between a “realistic to mildly illusionary positive appraisal” versus an “overly (delusionally) positive appraisal tendency” may not be straightforward to understand and e.g., concrete examples would be helpful. Another example is at the end of the first page of the introduction: “By focusing on appraisal contents, rather than on the cognitive processes that can generate these contents (as in PASS-process), the PASS-content scale more directly targets the PAS construct.” This sounds very abstract and also here an example or a more concrete writing style would help. A final example I would like to highlight comes from the discussion, where the authors speculate about the potential underlying processes PASS-process versus PASS-contents instruments target (i.e., unconscious vs. conscious). I think I understood the general idea, but it still may be helpful to critically re-check this section as it comes a bit out of the blue.

➔ REPLY: We are grateful to the reviewer for pointing out shortcomings in readability. In addition to the clarifications made in response to their above comments, we have worked on the manuscript in various places, in particular by giving more examples, and hope that the reviewer will judge the new version clearly more readable.

Concerning the reviewer’s concrete examples: The second paragraph of the introduction on the PAS construct has been extended with examples, and we have added a schematic graph (new Figure 1) that illustrates the idea of PAS, hoping to thereby facilitate comprehension. See P4, L10 ff. The paragraph at the end of the first page of the introduction now contains a link to the discussion (P5, L23: “See discussion for a more in-depth distinction.”), and in the corresponding paragraph of the discussion (P18, L17ff), we have added examples of positive (re)appraisal processes and positive appraisal contents from the respective questionnaires. This is also the paragraph

speaking about unconscious vs. conscious processes. We have simplified this discussion, as we felt that it was too complex and not essential to the paper.

This point also relates to some more general issues, e.g., I found out very late that the included moderators of the first two studies were selected based on previous analyses of the corresponding data sets. In addition, I think the analyses could be presented clearer, including the results that are presented in Table 1 and 2, e.g., I do not understand what the two columns within each column represent and it seems that the order of the columns are not presented in a chronological manner (which is fine but this might have a reason?). In addition, I am not sure whether I understand the general analytical set-up - the analyses seem to merely include predictors assessed at baseline and therefore represent a “prediction from baseline” analysis and not a crossed-legged analysis which would also control for certain variables from other time-point than baseline (and see point 3)?

→ REPLY: We believe there is a misunderstanding. The moderators (covariates) of the MARP and LORA studies were included as part of the same analysis plan, not in previous analyses. Covariate selection was done following the technical procedures and criteria established in previous work (Veer et al., 2023), but as part of the current MARP and LORA analyses, not as part of any previous work. To avoid further misunderstandings, this is now pointed out explicitly on P23, L29: “Covariate selection followed the technical procedures and criteria used in the initial studies (Bögemann et al., 2023; Veer et al., 2021; Zerban et al., 2023), applied to the present data”.

Regarding Tables 1 and 2, we see that our juxtaposition of the two separate prediction analyses (using either PASS-content or PASS-process as predictors) as sub-columns within a common column is confusing. We believe it is still good to have these side by side, such that the size of the beta estimates can be easily compared. In this way, the reader can immediately appreciate the generally higher estimates for PASS-content than for PASS-process as predictor. To prevent any confusion, we now use a color code, where the subcolumns showing the results for the analyses with PASS-content as predictor are colored beige and the subcolumns showing the PASS-process analyses are blue. PASS-content and PASS-process in the left columns showing all baseline variables are color-coded accordingly, and the legend explains the color code.

The order of the columns in terms of which time point is used as predictor and which interval as outcome is indeed not chronological. The most important analysis concerns the prediction of the entire (long) B0-B2 outcome interval by the B0 baseline variables (3.7 years in MARP, 3 years in LORA). This analysis is also reported first in the text and therefore placed first in the tables. All other analyses / columns only complement this major result, showing that predictions are also obtained when analyzing the shorter sub-intervals, while these do not add important information, e.g., they do not indicate much stronger predictions across shorter intervals. Hence, the tables in their present set-up and the text both convey the information of high temporal stability of the predictions. We would therefore

prefer to keep the current order of columns.

We are not entirely sure which additional analyses the reviewer would like to see and how these could further strengthen or question our overall picture of very stable prediction across long time intervals. However, we are happy to discuss and provide additional calculations, if the reviewer could kindly explain their request in more detail.

7. It would be interesting to also link the present data with some mental health data, which should be available in case I understood the methods section correctly, e.g., testing whether different operationalizations of appraisals (i.e., process vs. content) are related with depression and anxiety scores / predict these scores.

→ REPLY: Please note that the SR score IS a mental health outcome. It uses the sum score of the General Health Questionnaire (GHQ-28, in MARP and LORA) or Patient Health Questionnaire Anxiety and Depression Scale (PHQ-ADS, in the RCT), which assesses symptoms related to anxiety, depression, and in the case of the GHQ also psychosomatic signs, and social dysfunction. The only difference to an analysis simply trying to predict the GHQ or PHQ-ADS is that we here correct for stressor exposure, by integrating the GHQ or PHQ-ADS and the exposure data into the SR score. This is done in the intention to obtain a resilience-related outcome. We do not have a research question in this paper related to the prediction of a mere mental health score (raw GHQ or PHQ-ADS values).

Perhaps we also misinterpreted the reviewer's question. Could it be that the reviewer would like to see whether there are differential effects of the two PASS instruments on the depression or anxiety subscales of the GHQ/ PHQ-ADS (that is, the SR scores that could be built not using the entire GHQ/ PHQ-ADS but only its subscales)? We are happy to provide further analyses if the reviewer could kindly clarify.

Please also note that we do not have a prediction that appraisal processes vs. contents affect dimensions of psychopathology differently.

8. I may have overlooked this information but does the OSF project include a data codebook explaining all the variables included in the corresponding data files? The README file included some general information and I could not find anything in the other files?

→ REPLY: We have uploaded a codebook explaining the variables in each dataset. We have also updated the README file.

Reviewer #1 (Remarks on code availability):

Due to my upcoming holidays I was not able to review the code in much detail / repeat and check the analyses. I had a look at the files and in case I understood everything correctly there is no codebook describing the various variables in the data files.

→ REPLY: We have now provided a codebook, please also see our reply to comment #8.

Reviewer #2 (Remarks to the Author):

The paper includes several studies testing the potential causal effects of PAS on stress reactivity. It concerns quite a few impressive sets of data and the evidence for the potential causal role of PAS in resilience is convincing, which might have interesting potential for future studies on the prevention of exaggerated stress related responses.

The paper is quite an information rich paper, that is it includes a lot of information as it describes several (large) studies. Because data of three studies is used, a lot of information is also not provided, in order to make it fit in one paper, which makes it slightly rough to read and hard to wrap your head around. I think the paper makes a contribution to the field of psychology and might lead to more studies on this topic to further out insight in the concept of PAS and the role it could play in the stress resilience.

→ REPLY: We thank the reviewer for their encouraging comments. We indeed struggled with trying to fit the information about three studies into one paper. Reviewer 1 also requested further clarifications at several points in the manuscript. We invite the reviewer to read the revised version of the manuscript, and hope they will find it easier to follow. Unfortunately, the revisions have also made the manuscript longer, now exceeding the initial word limit.

I have a few questions that I would like the authors to respond to or / and address:

Minor question: why was in the MARP study a inclusion criterion used of three prior negative life events, specifically 3 (why not 1, or 2?), a rationale would be helpful to understand why this number was chosen.

→ REPLY: Literature on stress inoculation (e.g., Seery et al., 2010) suggested that a moderate number of past life events (2-4 in their sample of on aver 50-year olds) may be promotive of mental health, while a number of life events corresponding to the sample's average (7.7 in their paper) constitutes a risk factor. In a population-based cross-sectional sample of individuals in our area available to us at the time (Gutenberg Brain Study, GBS; see also Petri-Romão et al., 2024), we had observed an average number of about 4 events in 18 to 20-year olds, which corresponded to our attempted inclusion age in MARP. We therefore started data collection in MARP with an inclusion criterion of 4 or more past life events, in order to thereby enrich our sample for risk. Very early in data collection, it became apparent that very few interested potential participants would fulfil this criterion, and we therefore amended our procedures to include participants with 3 or more events in their past. In a reanalysis of the GBS data set, now comprising n=134 participants aged 18-20 at inclusion, we find an average

number of past life events of 3.05 for this group, retrospectively confirming the adequacy of our adjusted inclusion criterion. In brief form, this is now stated on P21, L12.

When introducing the RESPOND-RCT Spain is introduced in the introduction, it is explained that the first intervention step includes an ACT approach, the second step seems to rely on CBT interventions. It makes me curious and it would be interesting to already mention perhaps whether ACT and CBT had a differential effect on PAS. At least this question raises in my head while reading about the study, and it would be nice to mention some findings in the introduction already (or perhaps in the discussion when going into options to modify PAS).

→ REPLY: The reviewer raises an important point regarding stepped-care programs. While these programs reflect real-world medical and psychological practice more closely, they can be challenging to analyze in detail. In our study, all participants in the intervention arm received an initial ACT- and mindfulness-based intervention called DWM. Those who continued to report significant psychological distress after this first step (approximately 75% of participants) were subsequently invited to take part in a CBT-based intervention, PM+. Thus, while all participants received a stepped-care intervention, the specific components varied, which is intrinsic to stepped-care design but complicates the comparative analysis of its parts.

We observed a significant increase in PAS scores by 2.5 points from baseline (T0) to post-DWM (T1), followed by an additional significant increase in PAS by 1.7 points from post-DWM (T1) to post-PM+ (T2). This pattern suggests that both interventions may positively impact PAS; however, the interpretation is limited by potential carry-over effects from DWM to PM+ and the high transition rate from DWM to PM+.

To better understand the distinct mechanisms by which ACT, mindfulness, or CBT may influence PAS, alternative study designs could be considered. For example, future studies might compare stepped-care versus stratified treatment delivery (as in Copas et al., 2015), which more closely reflects real-life conditions, though it may still introduce biases as treatments are delivered based on participant characteristics. Another approach could involve a three-arm study comparing the effects of DWM, PM+, and usual care, on PAS separately. This could provide insights for refining these interventions to more precisely target positive appraisal, potentially leading to greater improvements in mental health. This is now briefly mentioned in the discussion (P19, L36-40).

Results: Pass-content shows full mediation in the LORA study, only partial mediation in the MARP study. Doesn't that mean that perhaps both social support and PASS-content seem to be important to focus on with the eye on interventions?

→ REPLY: When we report the PASS-content effects (the central result of the paper) as well as the (partial or full) mediation of social support effects by PASS-content in MARP and LORA, we do not mean to imply by this that

only PASS-content, but not social support, might be a promising target for intervention. By contrast, we completely agree that targeting social support might be beneficial or also that targeting both social support and PASS-content at the same time could be a promising approach. We think that the value of our paper is to show evidence for a strong and proximal effect of PAS.

We have now modified the concluding paragraph as follows (P20, L30): "... our results highlight promising avenues for promoting resilience. Next to broad and multifaceted interventions that also target more distal resilience factors, like the program employed in our RCT, resilience might also be enhanced by interventions specifically targeting PAS as a key proximal factor. This may involve positive mindset interventions⁵⁹, social-psychological interventions⁶⁰, or dedicated positive reappraisal^{61,62} or bias modification⁶³ trainings. Another promising approach might be to simultaneously play on a more distal factor that is mediated in its effect by PAS, such as social support, and on PAS. ... Such mechanistically more specific resilience trainings may become important tools in the global fight against stress-related disorders."

PAS seems to be almost the opposite of repetitive negative thinking (RNT), also a well known risk factor of psychopathology and stress responses. I was wondering what the view of the researchers is on this and how they think it relates to RNT (how does it differentiate from it perhaps?).

→ REPLY: This is an interesting question. Unlike RNT instruments, the PAS questionnaires do not address the temporal dimension of thinking about stressors. (Typical RNT questionnaire items include wording like "keep going through my mind", "can't stop", "repeat", "get stuck", "replay".) Further, the PAS questionnaires do not address to what extent the thoughts in question interfere with, or promote, other mental or physical activities (e.g., the interference subscale of the RNTQ). Finally, the thought processes addressed by RNT instruments seem to mostly involve unwarranted and spontaneous thinking ("worries pop into my head", "I am easily distracted", "thoughts come to my mind without me wanting to"). While positive appraisal processes and contents may also be spontaneously and non-deliberately generated, at least some of the processes in the PASS-process instrument can be considered deliberate emotion regulation strategies (positive cognitive reappraisal). On this basis, we would expect a moderate negative correlation between RNT questionnaires and the PASS instruments. We have currently no data set to test this prediction. In the absence of empirical data and given the exceedance of word limits we are already struggling with, we would like to refrain from including a discussion of this topic in the manuscript and hope that this finds the reviewer's agreement.

Although the results include a subanalysis when testing the longitudinal association between PASS and stress reactivity with only the most stress-exposed individuals (p.8). This is encouraging as it seems to support the view that the level of exposure does not influence the found longitudinal relationship. Also in the discussion the authors argue that "PAS appears to

protect against various types of stressors” and “participants experienced qualitatively different exposure” (p.18). Although I agree with this statement, based on the data that has been provided, I do wonder whether it also generalizes to more extreme stressors including traumatic events. Looking at the descriptives, most stressors that have been reported include common and relatively “mild” stressors. Even though I believe also here the dimensional model applies, it remains to be tested. Especially since traumatic events can shake one’s beliefs about the self, others and the world drastically. I would like the authors to elaborate a bit more on this discussion on generalizability to more severe or clinical contexts.

→ REPLY: The reviewer raises an important question. We have no data sets at our hands yet that allow us to test whether PAS also protects against the mental health sequelae of potentially traumatic events or extreme life events. We believe that this is a highly important question for future work and now explicitly discuss this on P19, L15: “One important question for future work is whether also exerts PAS protective effects in populations exposed to more extreme stressors than present in the samples analyzed so far, including severe potentially traumatic events.”

Reviewer #2 (Remarks on code availability): I did not run the code but data seems available

Reviewer #3 (Remarks to the Author):

The authors present a compelling argument for the importance of Positive Appraisal Style (PAS) as a crucial factor in stress resilience. The authors combine data from three studies – two longitudinal observational studies (MARP, LORA) and a randomized controlled trial (RESPOND-RCT Spain) – to demonstrate the predictive validity, mediating role, and modifiability of PAS. The results are generally strong and well-aligned with existing literature on resilience, offering valuable insights into potential pathways for bolstering mental health in the face of adversity.

The key attributes of this paper are the following:

Prospective evidence for PAS as a resilience factor: The prospective association between PAS and lower stressor reactivity (SR) across extended periods in both MARP and LORA convincingly establishes PAS as a robust predictor of resilience. This builds upon and significantly extends prior work that primarily focused on cross-sectional or short-term prospective associations.

Demonstration of mediation: The authors demonstrate that PAS mediates the relationship between perceived social support and SR in both observational studies. Additionally, the finding that perceived good stress recovery mediates the PAS-SR link aligns with the proposed mechanism of optimized stress response regulation through positive appraisals. This provides further support for the theoretical framework presented by PASTOR.

Intervention-induced changes in PAS: The results from the RESPOND-RCT Spain are particularly noteworthy, showing that a multi-component intervention

targeting a broad range of resilience factors leads to increased PAS, which in turn mediates improvements in SR. This strongly suggests a causal role for PAS in resilience, highlighting its potential as a target for therapeutic interventions.

Comprehensive and methodologically rigorous approach: The authors utilize a comprehensive approach, combining data from multiple studies with diverse populations and stressors. The use of validated instruments, residualized SR scores to control for exposure differences, and sophisticated mediation analyses strengthens the robustness of their findings.

→ REPLY: We thank the reviewer for these encouraging comments.

Here are some more detailed issues to address upon r&r:

The manuscript could benefit from a more explicit discussion of the relationship between PAS and other theoretical models of resilience, particularly those emphasizing flexible coping and emotion regulation (e.g., Bonanno's work, emotion regulation flexibility). How does this relate to existing models? Grounding this work in the larger research area may be warranted.

→ REPLY: We happily accept the reviewer's suggestion to more explicitly discuss a potential relationship of PASTOR with regulatory flexibility approaches to resilience, which we had done only briefly in the previous manuscript version, given word limits. Subsequent to our brief statement in the original manuscript: "Yet another challenge will be to empirically compare and integrate PASTOR with theories emphasizing the role of flexible coping and emotion regulation for resilience (e.g. (Aldao et al., 2015; Bonanno et al., 2024; Cheng et al., 2014; Roelofs et al., 2023; Sheppes et al., 2015; Troy et al., 2023))", we now write on P20, L21: "Flexibility-based theories assume that coping or emotion regulation strategies are not by themselves adaptive or maladaptive but that the adaptive value of a regulatory strategy is determined by the situation or context in which it is used. It is thus important to be able to flexibly shift between strategies as a function of situational demands. From the perspective of PASTOR, a tendency for positive appraisal makes it less likely that the person is overwhelmed by a stressful situation, which in turn makes it easier for them to employ different coping strategies in a situation-adapted fashion. Thus, positive appraisal permits regulatory flexibility, and regulatory flexibility can be an aspect of optimized stress response regulation. However, PASTOR does not claim that regulatory flexibility is always and necessarily an element of resilient responding to stressors(Kalisch et al., 2015, 2024)." We hope that the editors will accept that we exceed word limits.

Please also note that we are currently working on both empirical and theoretical comparisons of PASTOR and flexibility theories, which will be published separately.

The discussion of methodological limitations related to self-report measures of PAS is insightful. However, the authors could expand on potential strategies to

address these limitations in future research, such as incorporating objective measures of appraisal (e.g., physiological, implicit cognitive tasks) alongside self-report. What types of measures are both scalable and more precise and mechanistic than self-report?

→ REPLY: We equally happily include a short discussion of potential more objective measures, also because we are currently working on developing a corresponding task. On P20, L11, we complement the existing statement “One major challenge for future work will be to complement the existing self-report instruments for PAS with more objective, task-based measures” with: “that assess biases in behavioral or physiological responses to threat-related stimuli implicitly. Such biases may become apparent in particular in reactions to stimuli that are ambiguous with respect to their threat value. A suitable task should manipulate threat values on PASTOR’s three key threat appraisal dimensions of cost/magnitude, probability, and coping potential.”

Specificity of the intervention: While the multi-component nature of the RESPOND-RCT intervention is acknowledged, the authors could further elaborate on the specific components and their potential mechanisms of action. This would help clarify how the intervention might have targeted different pathways leading to PAS enhancement.

→ REPLY: This reviewer comment parallels a comment by reviewer 1, who wondered how the intervention elements of promoting social support, problem-solving, and acting on values and being kind might enhance PAS and thereby resilience (their point 2). In the newly added theoretical supplementary material 1 referred to on P6, L16 of the introduction, we now discuss these examples. Please apologize that we do not do this for all potential components of the interventions, since these are too numerous. The theoretical principle in any case is always the same: A key claim of PASTOR is that any resilience factor will ultimately benefit resilience because it shapes the way one typically appraises stressors (potential threats to one’s goals or needs) in a more positive way, and this is eventually what helps people stay mentally healthy despite adversity (Kalisch et al., 2015). In that paper in section 4.2.4.2, we have given the explicit example of social support as a more “distal” resilience factor and have posited that a person who perceives themselves as being well supported will usually perceive stressors as less threatening, for instance because they know that their support network would provide them with coping resources if needed. We also there argue that social support may have the additional effect of reducing one’s actual stressor exposure, because helpers may take some burden off one’s shoulders (see Figure 4A in Kalisch et al., 2015). Critically, however, this effect is covered by our extensive stressor exposure monitoring and the inclusion of this variable into our outcome score (SR score), which corrects for individual differences in exposure. Provided such correction, the only remaining statistically visible effect of social support on resilience should be via appraisal, according to the theory (Figure 4B in Kalisch et al., 2015). The example illustrates why and how PAS is posited to be a “proximal” resilience

factor, that is, an immediate causal factor in the effect path towards resilience and why and how it is thought to integrate the effects of other resilience factors. Analogous arguments apply to other resilience factors than social support (Kalisch et al., 2015, 4.2.4.2 uses the examples of life history and genotype).

To address the other potential resilience factors proposed by reviewer 1 as being positively affected by the intervention in our third study in the paper: If one has better problem-solving techniques at one's disposal, this will a) shape one's appraisal of coping potential positively (enhancing PAS), and it will b) reduce stressor exposure because problems get solved more efficiently (being factored out via the SR score). If one learns to act more in agreement with one's values and is kinder, this will a) reduce the perception of one's own actions as threats to one's self-esteem (one of the strongest stressors for most people) and hence enhance PAS, and it may b) also increase or decrease actual stressor exposure (e.g., increase because one avoids social conflicts less) but this will in any case be factored out through the SR score. In sum, our theory predicts an effect of any intervention on resilience via PAS, no matter what the exact ingredients of the intervention are. We chose to discuss these questions in a supplement in order to have more space, given the word limits for the main text.

Alternative explanations for intervention effects: The manuscript could benefit from a more detailed discussion of potential alternative explanations for the observed intervention effects. For example, could changes in stressor exposure (as indicated by the group by time interaction effect on E) have directly contributed to SR improvements, independent of PAS? Addressing such possibilities would strengthen the causal inferences drawn from the RCT data.

→ REPLY: This question directly links the reviewer's preceding question and our reply to it, in that it is conceivable that the pathways targeted by the intervention might involve reductions in stressor exposure. This is also apparent from the cited interaction effect on E. Importantly, however, the SR score corrects for individual differences in stressor exposure (see P5, L24 ff. in the introduction), such that we can exclude that this may have led to SR improvements and thus be an alternative explanation for the intervention effect on resilience. Please also see P16 L19 in the report of the RCT results, where we constated: "The effect [on E] highlights the need to control for exposure differences also in randomized trials." This inherent property is an important advantage of our metric.

Theoretical implications: The manuscript could benefit from a more comprehensive discussion of the broader theoretical implications of the findings for understanding resilience processes. How does PAS interact with other known resilience factors? Does PAS primarily operate at the level of primary appraisal (initial assessment of threat) or secondary appraisal (evaluation of coping resources)? Addressing these questions would enhance the theoretical contribution of the study.

➔ **REPLY:** Like reviewers 1 and 2, reviewer 3 also requests a more comprehensive discussion of the theoretical implications. Space restrictions had kept us from a more in-depth discussion in the previous manuscript version, and we hope the editors will accept that we now exceed the word limit. We partly solve the space problem through the use of supplements that provide a more in-depth discussion.

Specifically to the reviewer's points: With our reply to the reviewer's question on the intervention elements, we have already addressed the topic of how PAS might interact with other resilience factors. See the modified introduction paragraphs starting "PASTOR also claims that PAS is an integrative resilience factor that mediates the effects of other ..." (P5, L42 ff.) and "The idea of PAS as an integrative and proximal mediator..." (P6, L13ff., with the associated supplement) as well as the modified discussion paragraph starting "Our present data also showed that..." (P19, L24ff.).

On P18, L20, where we refer to a second theoretical supplementary material 2, we address the question of primary vs. secondary appraisal. Briefly, PASTOR (Kalisch et al., 2015) posits that in the case of milder stressors, initial (primary) appraisals may sometimes be positive, such that the overall appraisal outcome is also positive. This is termed "positive situation classification" and is believed to partly originate from positive experiences with similar stressors in the past or the application of cultural stereotypes to such mild stressor situations. In the case of more severe stressors, most initial appraisals are however believed to be negative, reflecting a natural default setting of our aversive system, which favors a cautious approach to possible threats ("better safe than sorry"). In these cases, to produce positive appraisal outcomes, secondary re-appraisals are necessary, which may include an assessment of coping potential. Note that most of the items in the PASS-process questionnaire can be interpreted as reflecting such secondary appraisals. It is therefore possible that tendencies in generating secondary appraisals make a relevant contribution to positive appraisal style.

Reviewer #3 (Remarks on code availability):

N/A

References

Aldao, A., Sheppes, G., & Gross, J. J. (2015). Emotion Regulation Flexibility. *Cognitive Therapy and Research*, 39(3), 263–278.

<https://doi.org/10.1007/s10608-014-9662-4>

Bөгemann, S. A., Puhmann, L. M. C., Wackerhagen, C., Zerban, M., Riepenhausen, A., Köber, G., Yuen, K. S. L., Pooseh, S., Marciniak, M. A., Reppmann, Z., Uściłko, A., Weermeijer, J., Lenferink, D. B., Mituniewicz, J., Robak, N., Donner, N. C., Mestdagh, M., Verdonck, S., van Dick, R., ... Kalisch, R. (2023). Psychological Resilience Factors and Their Association With Weekly Stressor Reactivity During the COVID-19 Outbreak in Europe: Prospective

Longitudinal Study. *JMIR Mental Health*, 10, e46518.
<https://doi.org/10.2196/46518>

Bonanno, G. A., Chen, S., Bagrodia, R., & Galatzer-Levy, I. R. (2024). Resilience and Disaster: Flexible Adaptation in the Face of Uncertain Threat. *Annual Review of Psychology*, 75, 573–599. <https://doi.org/10.1146/annurev-psych-011123-024224>

Cheng, C., Lau, H.-P. B., & Chan, M.-P. S. (2014). Coping flexibility and psychological adjustment to stressful life changes: A meta-analytic review. *Psychological Bulletin*, 140(6), 1582–1607. <https://doi.org/10.1037/a0037913>

Chmitorz, A., Neumann, R. J., Kollmann, B., Ahrens, K. F., Öhlschläger, S., Goldbach, N., Weichert, D., Schick, A., Lutz, B., Plichta, M. M., Fiebach, C. J., Wessa, M., Kalisch, R., Tüscher, O., Lieb, K., & Reif, A. (2020). Longitudinal determination of resilience in humans to identify mechanisms of resilience to modern-life stressors: The longitudinal resilience assessment (LORA) study. *European Archives of Psychiatry and Clinical Neuroscience*, 271(6), Article 6. <https://doi.org/10.1007/s00406-020-01159-2>

Copas, A. J., Lewis, J. J., Thompson, J. A., Davey, C., Baio, G., & Hargreaves, J. R. (2015). Designing a stepped wedge trial: Three main designs, carry-over effects and randomisation approaches. *Trials*, 16(1), 352. <https://doi.org/10.1186/s13063-015-0842-7>

Kalisch, R., Köber, G., Binder, H., Ahrens, K. F., Basten, U., Chmitorz, A., Choi, K. W., Fiebach, C. J., Goldbach, N., Neumann, R. J., Kampa, M., Kollmann, B., Lieb, K., Plichta, M. M., Reif, A., Schick, A., Sebastian, A., Walter, H., Wessa, M., ... Engen, H. (2021). The Frequent Stressor and Mental Health Monitoring-Paradigm: A Proposal for the Operationalization and Measurement of Resilience and the Identification of Resilience Processes in Longitudinal Observational Studies. *Frontiers in Psychology*, 12. <https://www.frontiersin.org/articles/10.3389/fpsyg.2021.710493>

Kalisch, R., Müller, M. B., & Tüscher, O. (2015). A conceptual framework for the neurobiological study of resilience. *Behavioral and Brain Sciences*, 38, e92. <https://doi.org/10.1017/S0140525X1400082X>

Kalisch, R., Russo, S. J., & Müller, M. B. (2024). Neurobiology and systems biology of stress resilience. *Physiological Reviews*. <https://doi.org/10.1152/physrev.00042.2023>

Kampa, M., Schick, A., Yuen, K., Sebastian, A., Chmitorz, A., Saase, V., Wessa, M., Tüscher, O., & Kalisch, R. (2018). *A Combined Behavioral and Neuroimaging Battery to Test Positive Appraisal Style Theory of Resilience in Longitudinal Studies* [Preprint]. Neuroscience. <https://doi.org/10.1101/470435>

Mediavilla, R., McGreevy, K. R., Felez-Nobrega, M., Monistrol-Mula, A., Bravo-Ortiz, M.-F., Bayón, C., Rodríguez-Vega, B., Nicaise, P., Delaire, A., Sijbrandij, M., Witteveen, A. B., Purgato, M., Barbui, C., Tedeschi, F., Melchior, M., van der Waerden, J., McDaid, D., Park, A.-L., Kalisch, R., ... Ayuso-Mateos, J. L. (2022).

Effectiveness of a stepped-care programme of internet-based psychological interventions for healthcare workers with psychological distress: Study protocol for the RESPOND healthcare workers randomised controlled trial. *DIGITAL HEALTH*, 8, 20552076221129084. <https://doi.org/10.1177/20552076221129084>

Petri-Romão, P., Engen, H., Rupanova, A., Puhlmann, L., Zerban, M., Neumann, R. J., Malyszau, A., Ahrens, K. F., Schick, A., Kollmann, B., Wessa, M., Walker, H., Plichta, M. M., Reif, A., Chmitorz, A., Tuescher, O., Basten, U., & Kalisch, R. (2024). Self-report assessment of Positive Appraisal Style (PAS): Development of a process-focused and a content-focused questionnaire for use in mental health and resilience research. *PloS One*, 19(2), e0295562. <https://doi.org/10.1371/journal.pone.0295562>

Roelofs, K., Bramson, B., & Toni, I. (2023). A neurocognitive theory of flexible emotion control: The role of the lateral frontal pole in emotion regulation. *Annals of the New York Academy of Sciences*, 1525(1), 28–40. <https://doi.org/10.1111/nyas.15003>

Seery, M. D., Holman, E. A., & Silver, R. C. (2010). Whatever does not kill us: Cumulative lifetime adversity, vulnerability, and resilience. *Journal of Personality and Social Psychology*, 99(6), 1025–1041. <https://doi.org/10.1037/a0021344>

Sheppes, G., Suri, G., & Gross, J. J. (2015). Emotion regulation and psychopathology. *Annual Review of Clinical Psychology*, 11, 379–405. <https://doi.org/10.1146/annurev-clinpsy-032814-112739>

Troy, A. S., Willroth, E. C., Shallcross, A. J., Giuliani, N. R., Gross, J. J., & Mauss, I. B. (2023). Psychological Resilience: An Affect-Regulation Framework. *Annual Review of Psychology*, 74, 547–576. <https://doi.org/10.1146/annurev-psych-020122-041854>

van der Heide, A., Dommershuijsen, L. J., Puhlmann, L. M. C., Kalisch, R., Bloem, B. R., Speckens, A. E. M., & Helmich, R. C. (2024a). Predictors of stress resilience in Parkinson's disease and associations with symptom progression. *Npj Parkinson's Disease*, 10(1), 1–10. <https://doi.org/10.1038/s41531-024-00692-4>

van der Heide, A., Dommershuijsen, L. J., Puhlmann, L. M. C., Kalisch, R., Bloem, B. R., Speckens, A. E. M., & Helmich, R. C. (2024b). Predictors of stress resilience in Parkinson's disease and associations with symptom progression. *NPJ Parkinson's Disease*, 10(1), 81. <https://doi.org/10.1038/s41531-024-00692-4>

Veer, I. M., Riepenhausen, A., Zerban, M., Wackerhagen, C., Puhlmann, L. M. C., Engen, H., Köber, G., Bögemann, S. A., Weermeijer, J., Uściłko, A., Mor, N., Marciniak, M. A., Askelund, A. D., Al-Kamel, A., Ayash, S., Barsuola, G., Bartkute-Norkuniene, V., Battaglia, S., Bobko, Y., ... Kalisch, R. (2021). Psycho-social factors associated with mental resilience in the Corona lockdown. *Translational Psychiatry*, 11(1), 67. <https://doi.org/10.1038/s41398-020-01150-4>

Zerban, M., Puhlmann, L. M. C., Lassri, D., Fonagy, P., Montague, P. R., Kiselnikova, N., Lorenzini, N., Desatnik, A., Kalisch, R., & Nolte, T. (2023). What helps the helpers? Resilience and risk factors for general and profession-specific

mental health problems in psychotherapists during the COVID-19 pandemic.
Frontiers in Psychology, 14, 1272199.
<https://doi.org/10.3389/fpsyg.2023.1272199>

Positive appraisal style predicts long-term stress resilience and mediates the effect of a pro-resilience intervention

Response to reviewers

Reviewer #1 (Remarks to the Author):

I would like to thank the authors for their hard work on the manuscript and for their thorough responses to the comments. As highlighted by Reviewer 2, this paper is rich in information and analyses. The revisions have significantly enhanced the quality of the work, and I believe this manuscript will make a valuable contribution to the literature.

Thank you very much for your thorough revision of our work.

Additional comments Reviewer #1 on Reviewer #2 rebuttal:

I was not able to locate one reply: the comment starting with “When introducing the RESPOND-RCT, Spain is introduced...” — it seems the authors did not copy-paste their addition into their reply, and the highlighted sections (P19, L36–40) remain unchanged.

Thank you very much for noticing this. The page number given in the last reply was wrong. These changes were introduced in page *number 20* (not 19), lines 36-41. The section reads:

Finally, it could be interesting to explore the differential contributions of the DWM and PM+ components of our RCT to PAS improvements, to thus identify the most efficient approach for shorter interventions and to better understand the relationships of their constitutive elements with appraisal style. Such mechanistically more specific resilience trainings may become an important tool in the global fight against stress-related disorders.

In the new version of the manuscript, this section remains unchanged. You may find it in the last paragraph of the Discussion (page 19 in the document with no tracked changes).

Regarding the comment beginning with “Although the results include a subanalysis when testing the longitudinal association between PASS and stress reactivity with...”, the reply is generally fine. However, it seems the authors were invited to elaborate a bit more, and their addition is currently rather brief — for example, they could cite studies that have already applied appraisal interventions with traumatized individuals. This section could be extended a bit further.

Thank you very much. This was our addition in the previous version of the manuscript:

One important question for future work is whether also exerts PAS protective effects in populations exposed to more extreme stressors than present in the samples analyzed so far, including severe potentially traumatic events.

To this we have now added the information:

This is conceivable given the substantial evidence supporting the role of stressor appraisal in the development of posttraumatic stress disorder (PTSD) symptoms (e.g., Gómez de la Cuesta et al., 2019, European Journal of Psychotraumatology, doi: 10.1080/20008198.2019.1620084).

Reviewer #2 (Remarks to the Author):

Thank you for your detailed responses to my comments and questions. Although I think that the authors have adequately addressed my questions, I do not find the manuscript with track changes, and it is therefore difficult to assess the extent to which all comments (including those of the other reviewers) have been addressed in the manuscript itself. Especially since this is still a very information-rich article that is not easy to read, I would like the authors to upload the manuscript with track changes for reasons of efficiency. If the authors have done this (I can't seem to find it), I would be happy to receive it from the editorial office, and then I can judge whether I'm satisfied with the revision.

Thank you very much!

We appreciate all the observations made by the Reviewer 2 during revision 1. We also appreciate that Reviewer 1 went through our response to Reviewer 2.

Reviewer #2 (Remarks on code availability):

will fill this out later

Reviewer #3 (Remarks on code availability):

I am satisfied that the authors have addressed my key concerns. Any lingering concerns are more stylistic than substantive.

Thank you very much for your revisions. We have also revised the style of our manuscript to make more accessible.